# Sub-Linear Memory: How to Make Performers SLiM

**Valerii Likhosherstov**
University of Cambridge
vl304@cam.ac.uk

**Krzysztof Choromanski**
Google Brain & Columbia University

**Jared Davis**
DeepMind & Stanford University

**Xingyou Song**
Google Brain

**Adrian Weller**
University of Cambridge & Alan Turing Institute

## Abstract

Transformer architectures have become very popular yet the original implementation requires $O(L^2)$ in serial time and memory as functions of input length $L$. Recent works proposed various linear self-attention mechanisms, scaling only as $O(L)$ for serial computation. We conduct a thorough complexity analysis of *Performers*, a class which includes most recent linear Transformer mechanisms. We note a remarkable computational flexibility: the gradient computation can be performed **with no approximations** using **sublinear memory** as a function of $L$ (in addition to negligible storage for the input sequence), at a cost of greater time complexity in the parallel setting. In the extreme case, a Performer consumes **only** $O(1)$ **memory**, and still requires $O(L)$ time. Due to complete backward-compatibility, this discovered time-memory tradeoff can be used for fine-tuning on low-memory devices in a decentralized fashion without any server computations.

## 1 Introduction

The Transformer architecture [38] has changed the landscape of deep learning for sequential data. A computational advantage of Transformers over conventional methods such as recurrent neural networks (RNNs) [17, 9] is parallelization over the sequence dimension, meaning that the training speed can be increased by simply using more compute resources. However, this parallel-friendly structure of self-attention comes at a cost of quadratic $\Theta(L^2)$ time and memory complexity, where $L$ is the length of the Transformer's input sequence.

A recent line of work aimed to address this restriction, using either structured sparsity [8], truncated back-propagation [12], clustering [20, 31] or linear attention methods [18, 10, 11, 33, 23]. For a detailed overview of efficient Transformers, see [37]. We refer to the family of linear attention architectures as *Performers* (also known as *Linear Transformers*), following [11], since their generic kernel formulation covers all the aforementioned linear attention methods. Performers reduce time and memory complexity to linear $O(L)$ and can provably approximate conventional quadratic Transformers [11], demonstrating strong performance in a systematic comparison of efficient Transformers [36].

This recent trend of feeding longer sequences into Transformers, coupled with the use of deeper models, introduces new challenges for researchers and practitioners. Whereas conventional Transformer setups benefit from large-batch optimization [42], long sequence modelling necessitates smaller batch sizes in order to fit the model into memory. For example, recently proposed efficient Transformers

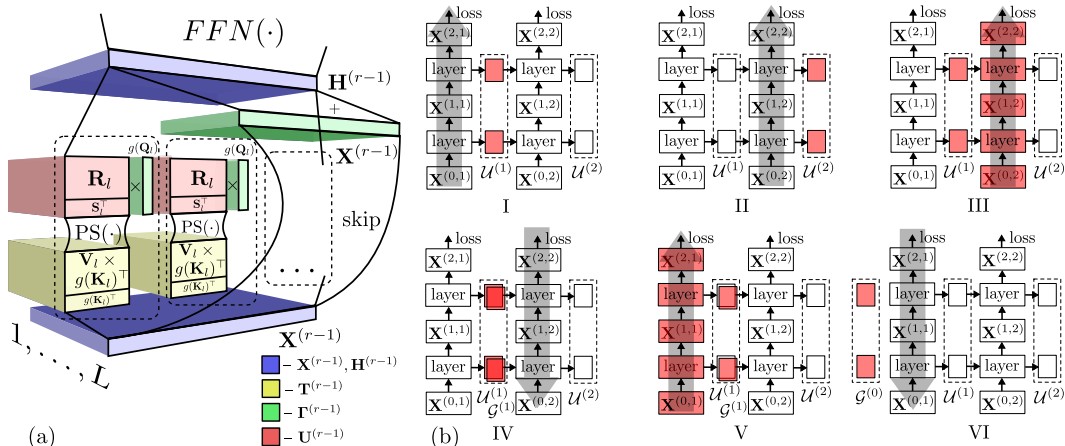

Figure 1: **(a)** $r$th layer and its decomposition into $\mathbf{T}^{(r-1)}, \mathbf{\Gamma}^{(r-1)}, \mathbf{U}^{(r-1)}$. **(b)** Illustration of Algorithm 1 when $r = n = 2$. Red color indicates objects stored in memory. I-II) forward passes for $n = 1, 2$ respectively, only the loss value and $\mathcal{U}^{(n)}$ are stored. III) backward pass start, forward computation through the slice $n = 2$ to build symbolic $\Phi^{(2)}$ and update $\mathcal{U}^{(2)} \rightarrow \mathcal{U}^{(1)}$. IV) back-propagation through $\Phi^{(2)}$ to find $\nabla_{\theta^{(2)}}\mathcal{L}$ and $\mathcal{G}^{(1)}$. V,VI) the same backward iteration for $n = 1$.

operating on long sequences used moderately small batch sizes of 1-8 instances [20, 18, 11]. Aiming to use larger batch sizes, practitioners introduced various tricks – e.g. [28] introduced *gradient accumulation* (included in the popular Fairseq library, [27]), which splits the batch into smaller chunks which are evaluated sequentially, then the resulting batch gradient is accumulated.

Gradient accumulation allows to decrease memory usage at the cost of longer time, but it can only be applied when the batch size is bigger than 1. In this paper, we are discussing a situation when even a batch size of 1 is prohibitive, while longer processing times are affordable e.g. when fine-tuning a pretrained Transformer on low-memory devices (e.g. smartphones, embedded devices or microcontrollers) on the client-generated data without additional server computations. Heuristics, such as chunking the input into independent subsegments or truncated back-propagation [12], limit gradient propagation across the whole input, and, consequently, impair long-context learning.

We propose a solution based on the analysis of Performers. We discover a remarkable property: even for batch size of 1, a user can decrease memory consumption at the cost of smaller parallel bandwidth of the model. Notably, **no approximations are introduced, so the obtained gradient is correct and backward-compatible**. Our proposed long-sequence training algorithm can be used for training or fine-tuning on a low-memory device, thus contributing towards decentralized and democratized deep learning. The algorithm has the following advantages:

1. The integer parameter $C, 1 \leq C \leq L$, controls a tradeoff between the memory, scaling as $O(C)$ in addition to a negligible input sequence storage, and parallel running time, scaling as $O((L/C)\log C)$. When $C = 1$, **the algorithm consumes as much memory as if a single token were fed into Performer**, plus a small fully characterized addition.

2. For any $C$, **the algorithm requires as many floating point operations (FLOPs) as two standard forward and one backward pass** plus a small addition.

We evaluate the proposed tradeoff empirically, and confirm backward-compatibility for the synthetic Copying Task and language modelling on Penn Treebank [25] and Enwik8 [24] datasets.[1]

---

[1]Code: `https://github.com/google-research/google-research/tree/master/performer/models/slim_performer`.

## 2 Background: linear self-attention and Performer

We commence by defining *linear self-attention* [18, 11, 33, 23]. Consider a sequence of length $L$ and three matrices: *queries* $\mathbf{Q} \in \mathbb{R}^{L \times d}$, *keys* $\mathbf{K} \in \mathbb{R}^{L \times d}$ and *values* $\mathbf{V} \in \mathbb{R}^{L \times d}$. Then linear self-attention is defined as a functional producing $\mathbf{Y} = \mathrm{Att}(\mathbf{Q}, \mathbf{K}, \mathbf{V}) \in \mathbb{R}^{L \times d}$,

$$\forall l \in \{1, \ldots, L\}: \mathbf{Y}_l = \frac{\sum_{l'=1}^{l} \mathbf{V}_{l'} \cdot (g(\mathbf{K}_{l'})^\top g(\mathbf{Q}_l))}{\sum_{l'=1}^{l} g(\mathbf{K}_{l'})^\top g(\mathbf{Q}_l)} = \frac{(\sum_{l'=1}^{l} \mathbf{V}_{l'} \times g(\mathbf{K}_{l'})^\top) \times g(\mathbf{Q}_l)}{(\sum_{l'=1}^{l} g(\mathbf{K}_{l'}))^\top g(\mathbf{Q}_l)}, \quad (1)$$

where by $\mathbf{Z}_l \in \mathbb{R}^{d_2 \times \cdots}$ we denote slice $\mathbf{Z}_{l,:,\ldots,:}$ of $\mathbf{Z} \in \mathbb{R}^{d_1 \times d_2 \times \cdots}$. $g: \mathbb{R}^d \to \mathbb{R}_+^M$ s a positive mapping for a stable division in (1). $M$ is typically much smaller than $L$, e.g. $M = d$.

For needs of autoregressive generative modelling, when each element depends only on previous elements of the sequence [38], $\mathbf{Y}_l$ only depends on inputs at indices $\{1, \ldots, l\}$. Self-attention of type (1) was proposed for processing sequences of the long length $L$, since the original self-attention from [38] scales as $O(L^2)$. The second transition in (1), which is due to associativity of matrix multiplication, suggests an algorithm to compute linear self-attention efficiently in subquadratic time.

For tensors $\mathbf{Z}^{(1)}, \ldots, \mathbf{Z}^{(n)}$ of the same shape, let $\mathbf{Z} = (\mathbf{Z}^{(i)})_{i=1}^n$ be a tensor such that for all $1 \leq i \leq n$, $\mathbf{Z}_{i,:,\ldots,:} = \mathbf{Z}^{(i)}$. By $\mathbf{R} \in \mathbb{R}^{L \times d \times M}$, $\mathbf{S} \in \mathbb{R}^{L \times M}$ denote a tensor and a matrix such that

$$\mathbf{R} = \mathrm{PS}((\mathbf{V}_l \times g(\mathbf{K}_l)^\top)_{l=1}^L), \ \mathbf{S} = \mathrm{PS}((g(\mathbf{K}_l))_{l=1}^L), \quad (2)$$

where $\mathrm{PS}(\mathbf{Z}) = (\sum_{i'=1}^{i} \mathbf{Z}_{i'})_{i=1}^n$ is a *prefix sum* along the first dimension of $\mathbf{Z}$. Next, compute

$$\forall 1 \leq l \leq L: \mathbf{Y}_l = (\mathbf{R}_l \times g(\mathbf{Q}_l))/(\mathbf{S}_l^\top g(\mathbf{Q}_l)). \quad (3)$$

Depending on the prefix-sum algorithm used in (2), we can obtain different complexity estimates for linear self-attention. [18] propose to iterate through $l = 1, \ldots, L$ maintaining only current $\mathbf{R}_l, \mathbf{S}_l$, and compute and store the result $\mathbf{Y}_l$. This way, tensors $\mathbf{R}, \mathrm{PS}(\mathbf{R}) \in \mathbb{R}^{L \times d \times M}$ are not stored in memory, resulting in $O(L)$ time complexity and $O(L(d + M) + dM)$ memory complexity. [18] also propose a similar computation of gradients through (2-3); see Appendix C for more discussion.

Alternatively, [11] employ a parallel prefix-sum algorithm [21, 39], which, for a tensor $\mathbf{Z} \in \mathbb{R}^{L \times \cdots}$, finds $\mathrm{PS}(\mathbf{Z})$ in $O(\log L)$ parallel time and $O(L)$ memory. Applying this algorithm for $\mathrm{PS}(\mathbf{R})$, $\mathrm{PS}(\mathbf{S})$ and then computing (3) results in only $O(\log L)$ parallel time complexity and $O(LdM)$ memory consumption. A similar approach was proposed in the context of parallel RNNs [26] .

Linear self-attention is a part of *Performers* [11] – efficient Transformers. Below we outline Performer architecture [11] for language modelling, while our analysis can be applicable in broader setups.

Let $\mathbf{p} \in \Sigma^L$ be an input sequence of length $L$, where $\Sigma$ is a finite alphabet. By $\mathrm{emb}(\mathbf{p}_l, l) \in \mathbb{R}^{d_{model}}$, $1 \leq l \leq L$, denote a linear combination of the $\mathbf{p}_l$ token's learned embedding and positional embedding of $l$'s position (sinusoids with different frequencies, as in [38]). Then Performer is defined as a parametrized mapping from $\mathbf{X}^{(0)} = (\mathrm{emb}(\mathbf{p}_l, l))_{l=1}^L \in \mathbb{R}^{L \times d_{model}}$ into $\mathbf{X}^{(out)} \in \mathbb{R}^{L \times |\Sigma|}$ through a sequence of hidden representations $\mathbf{X}^{(1)}, \ldots, \mathbf{X}^{(s)} \in \mathbb{R}^{L \times d_{model}}$. For each $1 \leq r \leq s$, $\mathbf{X}^{(r)}$ is obtained from $\mathbf{X}^{(r-1)}$ by applying a *Performer layer* consisting of two parts. The first part is a *multi-head self-attention* which comprises of $k$ linear self-attentions applied in parallel ($d_{model} = kd$) plus a skip connection. The second part is a feedforward network applied independently to each element of the sequence. See a formal definition of the architecture in Appendix A.

For each $l, 1 \leq l \leq L - 1$, $\mathbf{X}_l^{(out)}$ are predicted logits of the next token $\mathbf{p}_{l+1}$. Let $\mathcal{L}_l(\mathbf{X}_l^{(out)})$ denote the cross-entropy loss with respect to $\mathbf{p}_{l+1}$, or zero when $l = L$. The loss is defined as

$$\mathcal{L} = (L-1)^{-1} \cdot (\mathcal{L}_1(\mathbf{X}_1^{(out)}) + \cdots + \mathcal{L}_L(\mathbf{X}_L^{(out)})). \quad (4)$$

## 3 Low-memory back-propagation algorithm

### 3.1 Compact notation for Performer

We rewrite transformations $\mathbf{X}^{(0)} \to \mathbf{X}^{(1)} \to \cdots \to \mathbf{X}^{(s)}$ in the following form. For each $1 \leq r \leq s$,

$$\mathbf{T}^{(r-1)}, \mathbf{\Gamma}^{(r-1)} = F^{(r)}(\mathbf{X}^{(r-1)}; \theta), \ \mathbf{U}^{(r-1)} = \mathrm{PS}(\mathbf{T}^{(r-1)}), \ \mathbf{X}^{(r)} = G^{(r)}(\mathbf{U}^{(r-1)}, \mathbf{\Gamma}^{(r-1)}; \theta). \quad (5)$$

**Algorithm 1** Low-memory forward-backward pass. See Algorithm 2 for updateProc. Compared to notation from the text, redundant indices are dropped and tensor names are reused here and in Algorithm 2.

---

**Input:** $\mathbf{p} \in \Sigma^L$,
$\quad\quad \theta \in \mathbb{R}^{n_{param}}$,
$\quad\quad C \in \mathbb{N}$.
**Output:** loss $\mathcal{L}$,
$\quad\quad$ gradient $\nabla_\theta \mathcal{L}$.
Set $\mathcal{L} := 0$;
Set $\mathcal{U} := \mathbf{0}_{s \times D_1}$;
**for** $n = 1$ **to** $N$ **do**
$\quad$ updateProc($n$, False);
**end for**
Set $\nabla_\theta \mathcal{L} := \mathbf{0}_{n_{param}}$;
Set $\mathcal{G} := \mathbf{0}_{s \times D_1}$;
**for** $n = N$ **to** $1$ **do**
$\quad$ updateProc($n$, True);
**end for**
**Return** $\mathcal{L}, \nabla_\theta \mathcal{L}$ .

**Algorithm 2** updateProc procedure.

---

**Input:** $n \in \mathbb{N}$, binary flag onBP .
**if** onBP **then** Initialize $\Phi := 0$; **end if**
$\mathbf{X} := (\text{emb}(\mathbf{p}_{A_n+l}, A_n + l))_{l=1}^{B_n}$;
**for** $r = 1$ **to** $s$ **do**
$\quad$ Compute $\mathbf{T}, \mathbf{\Gamma} := F^{(r)}(\mathbf{X}; \theta)$;
$\quad$ **if** onBP **then** Update $\mathcal{U}_r \mathrel{-}= \sum_{l=1}^{B_n} \mathbf{T}_l$; **end if**
$\quad$ Set $\mathbf{U} := \mathbf{1}_{B_n} \mathcal{U}_r^\top + \text{PS}(\mathbf{T})$, $\mathbf{X} := G^{(r)}(\mathbf{U}, \mathbf{\Gamma}; \theta)$;
$\quad$ **if** onBP **then**
$\quad\quad$ Update $\Phi \mathrel{+}= \mathcal{G}_r^\top \mathbf{U}_{B_n}$;
$\quad$ **else**
$\quad\quad$ Update $\mathcal{U}_r := \mathbf{U}_{B_n}$;
$\quad$ **end if**
**end for**
Set $\mathcal{L}^{(upd)} := \mathcal{L}^{(n)}(\mathbf{X}\mathbf{W}^{(out)} + \mathbf{b}^{(out)})$;
**if** onBP **then**
$\quad$ Update $\Phi \mathrel{+}= \mathcal{L}^{(upd)}$;
$\quad$ Compute $\nabla_\theta \Phi, \nabla_\mathcal{U} \Phi$ through auto-differentiation;
$\quad$ Update $\nabla_\theta \mathcal{L} \mathrel{+}= \nabla_\theta \Phi$, $\quad \mathcal{G} := \nabla_\mathcal{U} \Phi$;
**else**
$\quad$ Set $\mathcal{L} \mathrel{+}= \mathcal{L}^{(upd)}$;
**end if**

Here $\theta \in \mathbb{R}^{n_{param}}$ is a set of all parameters, $\mathbf{T}^{(r-1)}, \mathbf{U}^{(r-1)} \in \mathbf{R}^{L \times D_1}$ and $\mathbf{\Gamma}^{(r-1)} \in \mathbf{R}^{L \times D_2}$ are the following matrices (see Figure 1a for an illustration). **(a)** $\mathbf{T}^{(r-1)}$ is a matrix of representations which are passed into the prefix-sum operator. That is, for each $1 \leq l \leq L$, $\mathbf{T}_l^{(r-1)}$ is a concatenation of $g(\mathbf{K}_l)$ and flattened $\mathbf{V}_l \times g(\mathbf{K}_l)^\top$ for all attention heads computed at the $r$th step. Consequently, $D_1 = M(d+1)k$. **(b)** For each $1 \leq l \leq L$, $\mathbf{U}_l^{(r-1)}$ is a concatenation of all corresponding $\mathbf{S}_l$ and flattened $\mathbf{R}_l$ – results of the prefix sum (Eq. 2) inside each self-attention head at $r$th layer. **(c)** $\mathbf{\Gamma}^{(r-1)}$ is a matrix of representations which skip the prefix sum. For each $1 \leq l \leq L$, $\mathbf{\Gamma}_l^{(r-1)}$ is a concatenation of $\mathbf{X}_l^{(r-1)}$ and $g(\mathbf{Q}_l)$ for each attention head $1 \leq j \leq k$ (3). Therefore, $D_2 = Mk + d_{model}$.

$F^{(r)}$ and $G^{(r)}$ are functionals depending on a set of model's parameters $\theta$. That is, they take subsets of $\theta$ corresponding to $r$th layer weights. $F^{(r)}$ is responsible for constructing $\mathbf{T}^{(r-1)}$ and $\mathbf{\Gamma}^{(r-1)}$ – representations preceding prefix sum, while $G^{(r)}$ finalizes multi-head self-attention and includes the feed-forward block. Importantly, $F^{(r)}$ and $G^{(r)}$ are applied **rowwise**, i.e. (5) can be rewritten as

$$\forall 1 \leq l \leq L : \mathbf{T}_l^{(r-1)}, \mathbf{\Gamma}_l^{(r-1)} = F^{(r)}(\mathbf{X}_l^{(r-1)}; \theta), \quad \mathbf{X}_l^{(r)} = G^{(r)}(\mathbf{U}_l^{(r-1)}, \mathbf{\Gamma}_l^{(r-1)}; \theta). \quad (6)$$

Hence, **the only place where the signal is propagated across the sequence is prefix sum in (5)**.

The representation (5) encapsulates architecture details of Performer inside $\{F^{(1)}, G^{(1)}, \dots, F^{(s)}, G^{(s)}\}$. In fact, the representation (5) holds for various possible modifications, proposed in the literature. This includes, but is not limited by the different positioning of layer normalization [41, 38], adding a stabilizing gating mechanism [29], weight sharing across layers [22] or reversible Transformer layers [20].

**High-level description of the algorithm.** The algorithm, proposed in the remainder of the section, iterates over the sequence $1, \dots, L$ and only maintains a front of current prefix sum values, thus allowing a substantial memory improvement. The backward pass is implemented similarly: current prefix sums along with their gradients are maintained in a dynamic programming fashion, but the iteration proceeds in a backward direction.

### 3.2 Forward computation

Suppose the memory budget is not enough to perform a complete forward pass through Performer (Equation 5 for $r = 1, \dots, s$), because the input sequence length $L$ is too big. We show that instead

we can emulate the full forward computation under the memory needed for a forward pass through the input of length $C \leq L$, plus a small addition. $1 \leq C \leq L$ is arbitrary and user-defined.

Split matrices $\mathbf{X}^{(r)}, \mathbf{T}^{(r)}, \mathbf{\Gamma}^{(r)}, \mathbf{U}^{(r)}$, into $N$ slices of size at most $C$ along the vertical axis ($N = \lceil L/C \rceil$): for each $n, 1 \leq n \leq N$,

$$\mathbf{X}^{(r,n)} = (\mathbf{X}^{(r)}_{A_n+l})^{B_n}_{l=1} \in \mathbb{R}^{B_n \times d_{model}}, \quad \mathbf{T}^{(r,n)} = (\mathbf{T}^{(r)}_{A_n+l})^{B_n}_{l=1},$$

$$\mathbf{U}^{(r,n)} = (\mathbf{U}^{(r)}_{A_n+l})^{B_n}_{l=1} \in \mathbb{R}^{B_n \times D_1}, \quad \mathbf{\Gamma}^{(r,n)} = (\mathbf{\Gamma}^{(r)}_{A_n+l})^{B_n}_{l=1} \in \mathbb{R}^{B_n \times D_2},$$

where $A_n = (n-1)C$ and by $B_n$, $1 \leq n \leq N$, we denote the size of the $n$th slice: $B_u = C$ for $u < N$, $B_N \leq C$. Based on (5), we conclude that for each $1 \leq n \leq N$ and $1 \leq r \leq s$ it holds that

$$\mathbf{T}^{(r-1,n)}, \mathbf{\Gamma}^{(r-1,n)} = F^{(r)}(\mathbf{X}^{(r,n)}; \theta), \quad \mathbf{U}^{(r-1,n)} = \mathbf{1}_{B_n} \times (\mathbf{U}^{(r-1,n-1)}_{B_{n-1}})^\top \tag{7}$$

$$+ \mathrm{PS}(\mathbf{T}^{(r-1,n)}), \quad \mathbf{X}^{(r,n)} = G^{(r)}(\mathbf{U}^{(r-1,n)}, \mathbf{\Gamma}^{(r-1,n)}; \theta). \tag{8}$$

Here, $\mathbf{1}_{B_n} \in \mathbb{R}^{B_n}$ is a vector of $B_n$ ones and we denote $\mathbf{U}^{(r-1,0)}_{B_0} = \mathbf{0}_{D_1}$ (a vector of $D_1$ zeros).

Now, instead of iterating over $r = 1, \ldots s$ and computing (5) for the whole sequence at once, we **first iterate over** $n = 1, \ldots, N$ **and then iterate over** $r = 1, \ldots, s$ **in a nested loop** to compute (7-8). As can be deduced from the (7-8), we only need to maintain the current value of $(\mathbf{U}^{(r-1,n-1)}_{B_{n-1}})^s_{r=1} \in \mathbb{R}^{s \times D_1}$ in the outer iteration over $n$.

Denote $\mathcal{U}^{(n)} = (\mathbf{U}^{(r-1,n)}_{B_n})^s_{r=1} \in \mathbb{R}^{s \times D_1}$, $0 \leq n \leq N$. The memory-efficient algorithm for the forward pass is as follows. First, initialize $\mathcal{L} = 0$ and $\mathcal{U}^{(0)} = \mathbf{0}_{s \times D_1}$. Then, iterate over $n = 1, \ldots, N$ and maintain the current value of $\mathcal{U}^{(n-1)}$. During each iteration, compute $\mathbf{X}^{(0,n)} = (\mathrm{emb}(\mathbf{p}_{A_n+l}, A_n + l))^{B_n}_{l=1}$. Then iterate over $r = 1, \ldots, s$, where compute (7-8) and update $\mathcal{U}^{(n)}_r = \mathbf{U}^{(r-1,n)}_{B_n}$. Finally, compute $\mathbf{X}^{(out,n)} = \mathbf{X}^{(s,n)}\mathbf{W}^{(out)} + \mathbf{b}^{(out)}$ and update $\mathcal{L} \mathrel{+}= \mathcal{L}^{(n)}(\mathbf{X}^{(out,n)})$, where we denote $\mathcal{L}^{(n)}(\mathbf{X}^{(out,n)}) = (L-1)^{-1} \sum_{l=1}^{B_n} \mathcal{L}_{A_n+l}(\mathbf{X}^{(out,n)}_l)$.

By the end of the iteration over $n$, the correct loss value (4) is computed. As a result, the forward pass takes $O(L)$ serial time or $O((L/C) \log C)$ parallel time and consumes only $O(C)$ memory. This is in addition to the input sequence $\mathbf{p} \in \Sigma^L$ storage, which is $O(L)$ in principle, however the constant is negligibly small. For instance, if $\mathbf{p}$ is a flattened image or an ASCII text string, then it occupies precisely $L$ bytes in memory. The $\log C$ term in the parallel time complexity is due to the parallel prefix-sum algorithm taking logarithmic time, as discussed in Subsection 2.

## 3.3 Back-propagation and the final algorithm

The goal of a backward pass is to compute the gradient $\nabla_\theta \mathcal{L}$ of the loss function with respect to parameters $\theta$. One can just perform automatic differentiation [14] (implemented in Tensorflow [1] and Pytorch [30]) through the computation graph induced by the memory-efficient forward pass algorithm from Subsection 3.2. However, such a backward pass would need to store all intermediate tensors produced during the forward pass, resulting in $O(L)$ memory complexity as a function of $L$ and $C$. Instead, we propose a back-propagation algorithm which has the same time and memory complexity as the efficient forward pass.

Let $\theta^{(1)} = \cdots = \theta^{(N)} = \theta$ be results of a symbolic "identity operation" performed on $\theta$, so that for all $1 \leq n \leq N$, $\theta^{(n)}$ is used instead of $\theta$ in (7-8). Then the total gradient of $\theta$ has the form $\nabla_\theta \mathcal{L} = \nabla_{\theta^{(1)}} \mathcal{L} + \cdots + \nabla_{\theta^{(N)}} \mathcal{L}$. In Appendix B we derive an expression for $\nabla_{\theta^{(n)}} \mathcal{L}$, $1 \leq n \leq N$. Namely, denote $\mathcal{G}^{(n)} = \nabla_{\mathcal{U}^{(n)}} \mathcal{L}$, then $\nabla_{\theta^{(n)}} \mathcal{L} = \nabla_{\theta^{(n)}} \Phi^{(n)}(\theta^{(n)}, \mathcal{U}^{(n-1)}, \mathcal{G}^{(n)})$, where $\Phi^{(n)} : \mathbb{R}^{n_{param}} \times \mathbb{R}^{s \times D_1} \times \mathbb{R}^{s \times D_1} \to \mathbb{R}$,

$$\Phi^{(n)}(\theta^{(n)}, \mathcal{U}^{(n-1)}, \mathbf{Z}) = \mathcal{L}^{(n)}(\mathbf{X}^{(out,n)}) + \sum_{r=1}^s \mathbf{Z}_r^\top \mathcal{U}_r^{(n)}.$$

In $\Phi^{(n)}$'s definition, $\mathbf{X}^{(out,n)}$ and $\mathcal{U}^{(n)} = (\mathbf{U}^{(r-1,n)}_{B_n})^s_{r=1}$ are results of (7-8) iteration over $r = 1, \ldots, s$ with parameters $\theta = \theta^{(n)}$ and $(\mathbf{U}^{(r-1,n-1)}_{B_{n-1}})^s_{r=1}$ equal to $\Phi^{(n)}$'s second argument $\mathcal{U}^{(n-1)}$. Gradient $\nabla_{\theta^{(n)}} \Phi^{(n)}$ can be computed by automatic differentiation through $\Phi^{(n)}$.

An efficient way to compute and sum up all $\nabla_{\theta^{(n)}} \mathcal{L}$ is to iterate in a backward direction $n = N, \dots, 1$ and to maintain values of $\mathcal{U}^{(n)}, \mathcal{G}^{(n)}$. $\mathcal{U}^{(N)}$ is known after the end of the forward pass, and

$$\forall 1 \leq n \leq N : \quad \mathcal{U}^{(n-1)} = \mathcal{U}^{(n)} - \sum_{l=1}^{B_n} (\mathbf{T}_l^{(r-1,n)})_{r=1}^s. \tag{9}$$

Further, in Appendix B we show that $\mathcal{G}^{(N)} = \mathbf{0}_{r \times D_1}$ and, for each $1 \leq n \leq N$,

$$\mathcal{G}^{(n-1)} = \nabla_{\mathcal{U}^{(n-1)}} \Phi^{(n)}(\theta^{(n)}, \mathcal{U}^{(n-1)}, \mathcal{G}^{(n)}). \tag{10}$$

By a single auto-differentiation through $\Phi^{(n)}$ we can compute $\nabla_{\theta^{(n)}} \mathcal{L} = \nabla_{\theta^{(n)}} \Phi^{(n)}$ and the update (10). If $\mathbf{w}$ is some vector of length $B_n$ and $h$ is some scalar function of $\mathbf{v} = \mathrm{PS}(\mathbf{w})$, then for all $1 \leq l \leq B_n : \nabla_h \mathbf{w}_l = \sum_{l'=t}^{B_n} \nabla_h \mathbf{v}_{l'}$. In other words, the gradient through $\mathrm{PS}(\cdot)$ is another prefix sum computed backwards. Hence, auto-differentiation through $\Phi^{(n)}$ takes the same parallel time $O(\log C)$, serial time $O(C)$ and memory $O(C)$, as the forward computation of $\Phi^{(n)}$. Since during the whole back-propagation algorithm, we only store and update tensors $\mathcal{U}^{(n)}, \mathcal{G}^{(n)}$, whose size doesn't depend on $L$ and $C$, this results in total $O((L/C) \log C)$ parallel time, $O(L)$ serial time and $O(C)$ memory in addition to $\mathbf{p}$ storage. A full description of the forward-backward pass is presented in Algorithm 1. Figure 1b is an illustration of the algorithm.

### 3.4 Analysis of the running time and memory

As we have shown, Performer can be trained in parallel time $O((L/C) \log C)$ and $O(C)$ memory in addition to the input $\mathbf{p}$ storage. Hence, $C$ is a tradeoff parameter: when $C$ is maximal ($C = L$), the model is fully-parallelized, therefore resulting in the fastest execution. Whereas minimal $C = 1$ corresponds to step-by-step processing, i.e. a fully-sequential regime which doesn't benefit from parallelized computations on GPU or TPU, but consumes $O(1)$ memory as a function of $L$.

During the forward pass, Algorithm 1 requires as many total FLOPs as the naive forward pass through (7-8). As for the backward pass, for each $1 \leq n \leq N$, the forward pass through $n$'s slice is repeated for symbolic construction of $\Phi^{(n)}$ (see Algorithm 2), and then back-propagation is run through $\Phi^{(n)}$. In addition, a backward update of $\mathcal{U}^{(n)}$ (9) is computed, taking precisely $B_n s M(d+1)k$ "add" operations. Hence, we conclude that **Algorithm 1 requires as many FLOPs as two forward and one backward pass through (7-8) for the whole sequence p** plus $LsM(d+1)k = LsMd_{model} + LsMk$ FLOPs. To characterize this addition, assuming that typically $d_{ff}$ (dimension of the feedforward block) is $4d_{model}$ in practice, observe that applying dense Performer layers (11-15, Appendix A) alone requires $3Lsd_{model}^2 + 2Lsd_{model}d_{ff} = 11Lsd_{model}^2$ FLOPs. This is much bigger than $LsMd_{model} + LsMk$, since $M$ is much smaller than $d_{model}$ in practice [11, 18].

Since the back-propagation takes roughly 5 times more FLOPs than the forward pass [14], we conclude that **memory efficiency of Algorithm 1 results in a small constant-time increase in FLOPs**. FLOPs affect energy consumption [40], a crucial factor for on-device applications.

Further analysis of Algorithm 1 reveals that the $C = 1$ regime requires **as much memory as if Transformer were applied to a sequence of length 1** plus exactly $2sd_{model}(M+1)$ floats for storing $\mathcal{U}, \mathcal{G}$. For comparison, the subset of $\theta$ corresponding to dense layers in self-attention and feed-forward blocks (11-15), occupies $3sd_{model}^2 + 2sd_{model}d_{ff} = 11sd_{model}^2$ floats. Again, this is much bigger than $2sd_{model}(M+1)$, since $M$ is much smaller than $d_{model}$ in practice.

Table 1 shows comparison of the proposed algorithm with other architectures such as a conventional Transformer, recurrent neural networks (RNNs, 17, 9) and residual networks (e.g. Neural ODEs, 6).

## 4 Experiments

Our main contribution is a new low-memory gradient computation algorithm for the existing Performer architecture. Performers have very competitive performance among other methods for long sequence modelling [11, 18, 36]. Hence, in the experimental section, we aim to answer the following questions about using this algorithm in practice. **(a)**. Does the theoretical time-memory tradeoff, controlled by $C$, agree with benchmarks of time-memory for varied $C$? **(b)** In precise arithmetic, different values of $C$ lead to the same correct gradient $\nabla_\theta \mathcal{L}$. Does this hold in practice, when

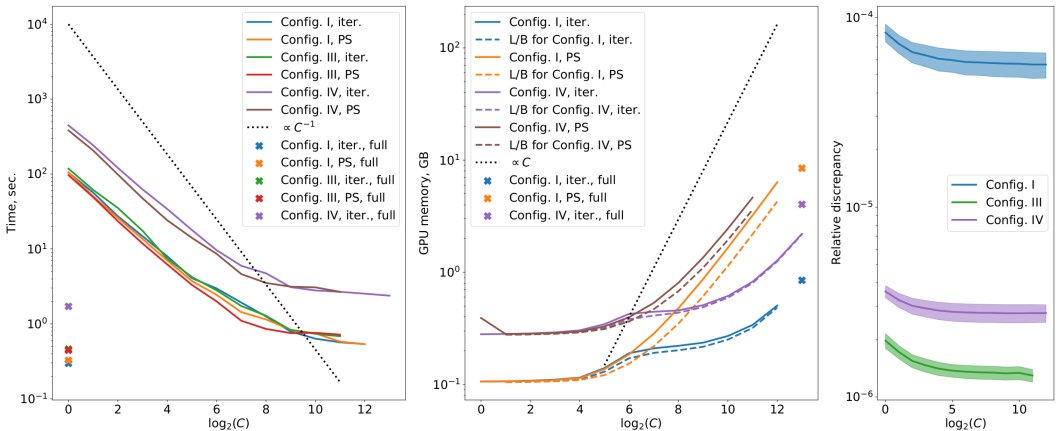

Figure 2: Benchmarks of Algorithm 1. All plots are averaged over 10 seeds. "iter." stands for iterative computation of (2-3), while "PS" is for explicit prefix sum computation in (2). We omit time and memory for big values of $C$ in "Config. IV, PS" and "Config. IV, full" setups, because these led to memory overflow. Analogous results for configuration II can be found in Appendix D. **(Left)** Time dependence on $C$. Crosses indicate horizontal time levels for corresponding full memory-inefficient methods. The dotted line indicates $\propto C^{-1}$ tangent in logarithmic scale. **(Middle)** Memory dependence on $C$. Again, crosses are for horizontal levels of full-sequence methods and the dotted line indicates $\propto C$ tangent. We do not report curves for config. III, because they completely match curves for config. IV, which is natural, since $s, d_{model}$ are the same for both configurations. "L/B" stands for a memory lower bound computed by processing input of length $C$. **(Right)** Relative gradient discrepancy as a function of $C$ and standard errors. Evaluated on random inputs over $\Sigma^L$.

finite-precision arithmetic is employed? **(c)** Can a model, pre-trained with a bigger $C$ (e.g. on a server), be fine-tuned with a smaller $C$ (e.g. on an embedded device)?

We address these questions below. We analyse 4 model configurations $(L, s, d_{model})$: I = (8192, 1, 1024), II = (1024, 3, 512), III = (4096, 3, 1024), IV = (16384, 3, 1024). In all configurations, we set $d_{ff} = 4d_{model}$, $k = d_{model}/64$ (number of heads). We set $M = d$ and employ $g(\mathbf{x}) = (\mathbf{x}_i^2)_{i=1}^d$ elementwise-quadratic feature mapping in (1), which we find to work well. In all experiments $\Sigma = \{0, \dots, 255\}$ and batch size is set to 1, i.e. we analyse a setup where gradient accumulation cannot be used to decrease memory, and therefore our algorithm is crucial. Our code is in PyTorch 1.7 [30]. To ensure that reproduction of experiments is accessible for a wider audience, we use a single NVIDIA Tesla P100 GPU with 16 GB memory for each experiment.

## 4.1 Empirical benchmarking of the tradeoff

We run Algorithm 1 for configurations I, III, IV and different powers of 2 as $C$. We use input strings sampled randomly from $\Sigma^L$. In order to characterize the time-memory tradeoff, we measure wall-clock time and peak GPU memory for a single gradient evaluation. We use the `torch.cuda.max_memory_allocated` function to report peak GPU memory.

As discussed in Section 2, there are two methods to compute (2-3): the first (iterative) method doesn't compute and store tensors (3) explicitly, resulting in smaller memory consumption at a cost of less parallelization, while the second one computes tensors (3) using the parallel prefix sum algorithm, therefore operating faster, but using more memory. The same methods can be applied for the memory-efficient algorithm when computing (7-8) updates. We implement and benchmark both methods as part of the algorithm. For the explicit prefix-sum method, we find that the `torch.cumsum` function works faster and consumes less memory than our custom implementation of the parallel prefix sum algorithm. We attribute this to hardware-optimized implementation of the native function, and use it in experiments. As for the iterative algorithm, we implement its "block" version, when, instead of iterating $l$ one-by-one, we iterate through blocks of small size (see details in Appendix

Table 1: Complexity for the back-propagation as a function of sequence length $L$ and the tradeoff parameter $C \leq L$. The indicated memory complexity is in addition to the input sequence $\mathbf{p}$ storage. The serial time complexity for Performer is reported for the version with iterative $\mathrm{PS}(\cdot)$ computation (as in [18]), while the parallel time is reported for the parallel prefix sum (as in [11]). For both methods, memory complexity is the same, but the constant is smaller for the iterative version.

| Model | Serial time | Parallel time | Memo-ry |
|---|---|---|---|
| RNN | $O(L)$ | $O(L)$ | $O(L)$ |
| Residual NN | $O(L)$ | $O(L)$ | $O(L)$ |
| Transformer | $O(L^2)$ | $O(\log L)$ | $O(L^2)$ |
| Performer | $O(L)$ | $O(\log L)$ | $O(L)$ |
| Our alg. | $O(L)$ | $O(\frac{L}{C}\log C)$ | $O(C)$ |
| Ours, $C=1$ | $O(L)$ | $O(L)$ | $\mathbf{O(1)}$ |

Table 2: Time per iteration (sec., averaged over 1000 iterations) and peak GPU memory (Gb). When using small values of $C$, we relate the remaining memory to a storage of parameters $\theta$ and optimizer's state.

| Setup, $L$, $C$ | Time | Memory |
|---|---|---|
| CT, 8192, full | 0.3008 | 0.938 |
| CT, 8192, 4096 | 0.5372 | 0.595 |
| CT, 8192, 2048 | 0.6002 | **0.436** |
| PTB, 1024, full | 0.1377 | 0.300 |
| PTB, 1024, 512 | 0.2526 | 0.257 |
| PTB, 1024, 256 | 0.3060 | **0.231** |
| ENW, 4096, full | 0.4598 | 1.513 |
| ENW, 4096, 2048 | 0.7922 | 1.085 |
| ENW, 4096, 1366 | 0.8654 | **0.909** |

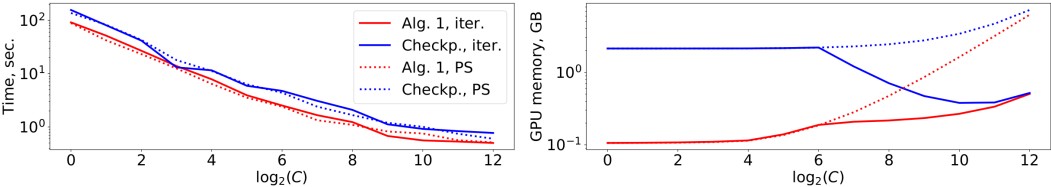

Figure 3: Algorithm 1 compared to checkpointing of $\{\mathcal{U}^{(n)}\}_{1 \leq n \leq N}$. Time and memory plots.

$C$). This way, the algorithm has a smaller constant in $O(L)$ time complexity and bigger constant in a "small" $O(dM)$ term of the memory complexity (assuming that $d, M \ll L$).

For a fixed $C$, in addition to reporting memory of Algorithm 1, we also report memory of the naive gradient computation run on a string of length $C$, sampled uniformly from $\Sigma^C$. This is to confirm that memory use of Algorithm 1 is just slightly above the full computation on the input of length $C$.

Results are reported in Figure 2 (left, middle). We observe significant improvements in memory compared to the full computation, as $C$ decreases. As $C$ converges to $2^0 = 1$, **the remaining memory can be attributed to storage of the model's parameters** $\theta$. Time follows two regimes: declining fast as $C$ grows (meaning that prefix sums are parallelized) and declining slower for big values of $C$ (meaning that the practical limit of parallelization is reached). Memory scales slower than $O(C)$, as $C$ increases. We attribute this to details of the PyTorch internal code. We find that iterative version of (2-3) works only slightly slower than prefix-sum version, while consuming much less memory. Finally, **Algorithm 1 consumes only slightly more memory in practice than the full method run on the input of length** $C < L$ ("L/B" plots on Figure 2 (middle)).

In addition, we compare Algorithm 1 with checkpointing [15]. By checkpointing in this context we understand storing $\{\mathcal{U}^{(n)}\}_{1 \leq n \leq N}$ during the forward pass and reusing them during the backward pass instead of recomputing. This results in a small FLOPs decrease since $\mathcal{U}^{(n)}$, while memory scales as $O(L/C)$. We use config. I for comparison (Figure 3). While not faster in practice, checkpointing consumes much more memory as $C$ decreases for both iterative and prefix-sum computation of (2-3).

## 4.2 Effects of finite-precision arithmetic

Since the iterative version of (2-3) results in a good balance between time and memory of Algorithm 1, we use it further. To quantify finite-precision effects, we plot *relative discrepancy* $\|\nabla_\theta^{(C)}\mathcal{L} - \nabla_\theta^{(full)}\mathcal{L}\|_2/\|\nabla_\theta^{(full)}\mathcal{L}\|_2$ between the gradient $\nabla_\theta^{(C)}$ produced by Algorithm 1, and the gradient

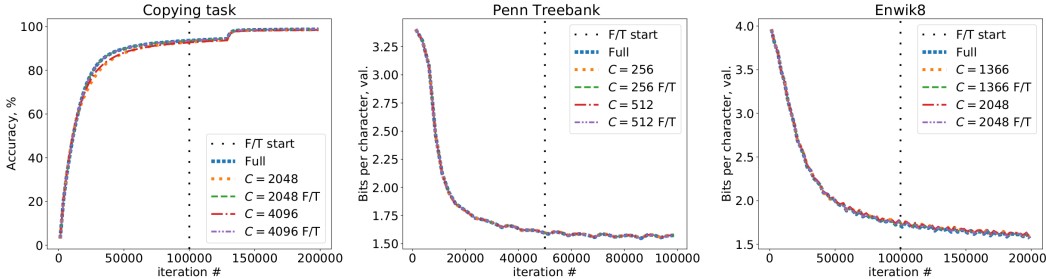

Figure 4: Learning curves for three language modelling setups. We report accuracy on a newly generated data samples for Copying task, and Bits Per Character (BPC) metric on validation examples for Penn Treebank and Enwik8. F/T stands for "fine-tuning". **All curves are roughly the same and result in nearly the same performance, confirming correctness and backward-compatibility** of gradients computed via memory-efficient Algorithm 1. Differences can be attributed to finite-precision arithmetic effects, accumulating over many iterations.

Table 3: Time per iteration (sec., averaged over 1000 iterations) and peak GPU memory (Gb). We also report memory required to store parameters $\theta$ only without any computations.

| Setup, $L$, $C$ | BPC, F/T | BPC, no F/T | Memory, params | F/T mem., Alg. 1 | F/T mem., full comp. | F/T time, Alg. 1 | F/T time, full alg. |
|---|---|---|---|---|---|---|---|
| PTB, 1024, 16 | **1.4263** | 1.6544 | 0.0661 | **0.1077** | 0.1834 | 0.8938 | 0.0426 |
| ENW 4096, 64 | **1.5642** | 1.6141 | 0.2610 | **0.4276** | 0.8049 | 1.1142 | 0.2045 |

$\nabla_\theta^{(full)}\mathcal{L}$ produced by full computation. Figure 2 shows results for randomly initialized models. We observe a small discrepancy of order $10^{-6}$–$10^{-4}$, confirming the correctness of Algorithm 1.

### 4.3 Training from scratch and fine-tuning

To confirm backward compatibility of Algorithm 1 during training, we run three language modelling setups: Copying Task (CT), symbol-level Penn Treebank (PTB) and Enwik8 (ENW).

For the CT, we follow the setup from [20, 18], sampling inputs as $0\omega0\omega$, where $\omega$ is drawn uniformly from $(\Sigma \setminus \{0\})^{L/2-1}$. In this setup, we only aggregate cross-entropy loss from the second half of the input, so the task is to reproduce the first half. We include the CT as a task where long-range signal is crucial, and the heuristic of "chunking" the input into segments would fail to solve the task.

We use model configurations I, II, III for the CT, PTB and ENW, resulting in sequence lengths $L = 8192, 1024, 4096$ respectively. For each setup, we compare training with full gradient computation, "fine-tuning" regime, when the first half of iterations is run using the full algorithm, and the second half is run using Algorithm 1 using various values of $C$. In addition, we include training from scratch equipped with memory-efficient gradient computation via Algorithm 1. Figure 4 demonstrates results: all methods result in almost same performance. **This confirms that memory-efficient gradient computation is backward-compatible during training**. Table 2 quantifies the memory savings and time tradeoff in all setups. Additional details and results, including bigger version of Figure 4, BPC for CT, train set performance, and curve differences in Figure 4, can be found in Appendix D.

### 4.4 One-shot fine-tuning under low memory

To analyze the scenario when model is pretrained on server and then fine-tuned (F/T) with a small $C$ on a low-memory device, we add the following experiment. We take a pretrained model from either PTB or ENW setup from Section 4.3 and subsample randomly 5000 examples from the corresponding validation set. We perform a one-step gradient descent with $0.01$ learning rate (tuned on other random subset) to minimize the loss computed on the first half of each sequence and evaluate Bits Per Character (BPC) on the second half. In this experiment, the first half of the sequence represents the data generated by user on the device, and the second half is a new data to be predicted. Fine-tuning

procedure, therefore, represents "personalization" of the model to the specific user (see Table 3). We observe BPC improvement without any server compute and a memory improvement compared to the full computation, while time ($\approx 1$ sec.) is less crucial, since fine-tuning can run in the background.

# 5    Related Work and Extensions

**Compatibility with other memory-optimization techniques.** Observe that the specification (5) is compatible with the reversible layer design from [20], when the sparse self-attention is replaced with the linear self-attention[2]. This can bring more memory savings, since one doesn't need to store the whole symbolic $\Phi^{(n)}$ during the backward pass. Checkpointing along the layer dimension [15, 7] can also be used to reduce the memory consumption for storing $\Phi^{(n)}$'s graph, though at the cost of a longer execution time. The gradient accumulation technique [28] is also compatible with Algorithm 1, i.e. one can combine both methods to "collapse" batch and sequence dimensions simultaneously. Moreover, our algorithm is compatible with distillation [32], since it can be run on a distilled model.

**Comparison with [18].** In [18], authors mention that a single self-attention block can be evaluated in $O(1)$ additional memory. However, during back-propagation, one still needs to store $L$ intermediate states, e.g. in the feedforward block. Hence, the full memory complexity is still $O(L)$. In contrast, our method optimizes memory consumption along the sequence dimension for the whole multilayer model.

**Extension to Transformers with dropout.** Dropout [34] is a popular regularization technique. It is used with Transformers when the train dataset is small enough to cause overfitting (e.g. it wasn't used with GPT-2, trained on a massive dataset). Our algorithm can be extended to stochastic computation graphs with dropout. For that, use separate random seeds to generate dropout masks for each slice $1 \leq n \leq N$, and reuse these seeds two times during the forward and backward pass through the $n$th slice.

# 6    Conclusion

We proposed an algorithm for memory-efficient back-propagation through a Performer. The algorithm reduces memory consumption along the sequence dimension, and can, therefore, be used for long-sequence training. The algorithm: (1) is completely backward-compatible, since it computes precise gradients and does not involve approximation, (2) does not require many additional computations, and (3) enables user control over the tradeoff between time and memory consumption.

**Limitations.** One limitation of this works is that the proposed memory improvements are traded off by longer running time. Another one is that these improvements rely on a specific feature of prefix sum signal propagation appearing in Performers and, also, in parallelized versions of RNNs [26]. Finally, we don't see an immediate extension of the proposed construction to the bidirectional self-attention case, used in Transformer encoders [38] and for masked language modelling [13]. The reason for that is that no notion of the prefix sum "front" can be introduced in this case.

**Negative societal impacts.** This work studies Performers – efficient Transformers. Transformers in general can have the following negative societal impacts: large carbon dioxide emissions [35], privacy and data leak vulnerabilities [5], bias and fairness issues and malicious misuse [4, 3].

# Acknowledgments

We thank Tom Weingarten and Tamas Sarlos for many fruitful discussions.

**Funding disclosure**. V. L. acknowledges support from the Cambridge Trust and DeepMind. V. L. was part-time employed by Google while a PhD student. A.W. acknowledges support from a Turing AI Fellowship under grant EP/V025379/1, The Alan Turing Institute, and the Leverhulme Trust via CFI.

---

[2]See e.g. `CausalFavor` class in https://github.com/ google/trax/blob/master/trax/layers/research/sparsity.py, which is compatible with the official Reformer code.

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
