## A Performer architecture details

We define the Performer architecture formally as follows. $\mathbf{X}^{(out)} = \mathbf{X}^{(s)}\mathbf{W}^{(out)} + \mathbf{b}^{(out)}$ and for each $1 \leq r \leq s$:

$$\mathbf{H}^{(r-1)} = \text{LN}(\text{MultiHead-Att}(\mathbf{X}^{(r-1)})) + \mathbf{X}^{(r-1)}, \tag{11}$$

$$\mathbf{X}^{(r)} = \text{LN}(\text{FFN}(\mathbf{H}^{(r-1)})) + \mathbf{H}^{(r-1)}, \text{ where} \tag{12}$$

$$\text{MultiHead-Att}(\overline{\mathbf{X}}) = [\mathbf{H}^{(1)} \dots \mathbf{H}^{(k)}], \tag{13}$$

$$\forall j \leq k : \mathbf{H}^{(j)} = \text{Att}(\overline{\mathbf{X}}\mathbf{W}_Q^{(j)}, \overline{\mathbf{X}}\mathbf{W}_K^{(j)}, \overline{\mathbf{X}}\mathbf{W}_V^{(j)}), \tag{14}$$

$$\text{FFN}(\overline{\mathbf{H}}) = \text{GeLU}(\overline{\mathbf{H}}\mathbf{W}^{(1)} + \mathbf{b}^{(1)})\mathbf{W}^{(2)} + \mathbf{b}^{(2)}. \tag{15}$$

Here $k$ is the number of attention heads ($d_{model} = kd$). $\mathbf{W}^{(out)} \in \mathbb{R}^{d_{model} \times |\Sigma|}$, $\mathbf{b}^{(out)} \in \mathbb{R}^{1 \times |\Sigma|}$, $\mathbf{W}^{(1)} \in \mathbb{R}^{d_{model} \times d_{ff}}$, $\mathbf{b}^{(1)} \in \mathbb{R}^{1 \times d_{ff}}$, $\mathbf{W}^{(2)} \in \mathbb{R}^{d_{ff} \times d_{model}}$, $\mathbf{b}^{(2)} \in \mathbb{R}^{1 \times d_{model}}$, $\mathbf{W}_Q^{(j)}, \mathbf{W}_K^{(j)}, \mathbf{W}_V^{(j)} \in \mathbb{R}^{d_{model} \times d}$ are trainable parameters (separate for each instance of MultiHead-Att, FFN), "+" is broadcasted rowwise when biases are added and LN is layer normalization [2], which is applied rowwise and depends on additional trainable parameters. GeLU denotes Gaussian error Linear Unit [16], which is applied elementwise.

## B Derivation of Gradient Expressions

$\theta^{(n)}$ doesn't affect terms $\mathcal{L}^{(1)}(\mathbf{X}^{(out,1)}), \dots, \mathcal{L}^{(n-1)}(\mathbf{X}^{(out,n)})$, so corresponding gradients are zero:

$$\nabla_{\theta^{(n)}}\mathcal{L} = \nabla_{\theta^{(n)}} \sum_{n'=n}^{N} \mathcal{L}^{(n')}(\mathbf{X}^{(out,n')}).$$

Similarly, $\mathcal{U}^{(n)}$ does not affect $\mathcal{L}^{(1)}, \dots, \mathcal{L}^{(n)}$, so

$$\mathcal{G}^{(n)} = \nabla_{\mathcal{U}^{(n)}}\mathcal{L} = \nabla_{\mathcal{U}^{(n)}} \sum_{n'=n+1}^{N} \mathcal{L}^{(n)}(\mathbf{X}^{(out,n')}).$$

In particular,

$$\mathcal{G}^{(N)} = \nabla_{\mathcal{U}^{(N)}}\mathcal{L} = \mathbf{0}_{r \times D_1}.$$

For all $1 \leq n < n' \leq N$, $\theta^{(n)}$ and $\mathcal{U}^{(n-1)}$ affect $\mathcal{L}^{(n')}$ only through $\mathcal{U}^{(n)}$, so according to the chain rule

$$\nabla_{\theta^{(n)}} \sum_{n'=n+1}^{N} \mathcal{L}^{(n')}(\mathbf{X}^{(out,n')}) = \sum_{r=1}^{s} \frac{\partial \mathcal{U}_r^{(n)}}{\partial \theta^{(n)}}^{\top} \times \nabla_{\mathcal{U}_r^{(n)}} \sum_{n'=n+1}^{N} \mathcal{L}^{(n')}(\mathbf{X}^{(out,n')})$$

$$= \sum_{r=1}^{s} \frac{\partial \mathcal{U}_r^{(n)}}{\partial \theta^{(n)}}^{\top} \times \nabla_{\mathcal{U}_r^{(n)}}\mathcal{L},$$

$$\forall 1 \leq r' \leq s : \nabla_{\mathcal{U}_{r'}^{(n-1)}} \sum_{n'=n+1}^{N} \mathcal{L}^{(n')}(\mathbf{X}^{(out,n')}) = \sum_{r=1}^{s} \frac{\partial \mathcal{U}_r^{(n)}}{\partial \mathcal{U}_{r'}^{(n-1)}}^{\top} \times \nabla_{\mathcal{U}_r^{(n)}} \sum_{n'=n+1}^{N} \mathcal{L}^{(n')}(\mathbf{X}^{(out,n')})$$

$$= \sum_{r=1}^{s} \frac{\partial \mathcal{U}_r^{(n)}}{\partial \mathcal{U}_{r'}^{(n-1)}}^{\top} \times \nabla_{\mathcal{U}_r^{(n)}}\mathcal{L},$$

where $\frac{\partial \square}{\partial \square}$ denotes Jacobian matrices. Further, for all $1 \leq r \leq s$:

$$\frac{\partial \mathcal{U}_r^{(n)}}{\partial \square}^{\top} \times \nabla_{\mathcal{U}_r^{(n)}}\mathcal{L} = \nabla_{\square}\left([\mathcal{U}_r^{(n)}]^{\top}\langle\langle\nabla_{\mathcal{U}_r^{(n)}}\mathcal{L}\rangle\rangle\right),$$

where $\square \in \{\theta^{(n)}\} \cup \{\mathcal{U}_{r'}^{(n-1)}\}_{1 \leq r' \leq s}$. $\langle\langle \cdot \rangle\rangle$ denotes a *stop-gradient* operator, i.e. gradients are not propagated inside brackets and the argument is considered as constant.

We conclude that

$$\nabla_{\theta^{(n)}} \mathcal{L} = \nabla_{\theta^{(n)}} \mathcal{L}^{(n)}(\mathbf{X}^{(out,n)}) + \nabla_{\theta^{(n)}} \sum_{n'=n+1}^{N} \mathcal{L}^{(n')}(\mathbf{X}^{(out,n')}) = \nabla_{\theta^{(n)}} \mathcal{L}^{(n)}(\mathbf{X}^{(out,n)})$$

$$+ \sum_{r=1}^{s} \frac{\partial \mathcal{U}_r^{(n)}}{\partial \theta^{(n)}}^{\top} \times \nabla_{\mathcal{U}_r^{(n)}} \mathcal{L}$$

$$= \nabla_{\theta^{(n)}} \left( \mathcal{L}^{(n)}(\mathbf{X}^{(out,n)}) + \sum_{r=1}^{s} [\mathcal{U}_r^{(n)}]^{\top} \langle\langle \nabla_{\mathcal{U}_r^{(n)}} \mathcal{L} \rangle\rangle \right) = \nabla_{\theta^{(n)}} \Phi^{(n)}(\theta^{(u)}, \mathcal{U}^{(n-1)}, \nabla_{\mathcal{U}^{(n)}} \mathcal{L})$$

$$= \nabla_{\theta^{(n)}} \Phi^{(n)}(\theta^{(u)}, \mathcal{U}^{(n-1)}, \mathcal{G}^{(n)}),$$

$$\forall 1 \le r' \le s : \mathcal{G}_{r'}^{(n-1)} = \nabla_{\mathcal{U}_{r'}^{(n-1)}} \mathcal{L} = \nabla_{\mathcal{U}_{r'}^{(n-1)}} \mathcal{L}^{(n)}(\mathbf{X}^{(out,n)}) + \nabla_{\mathcal{U}_{r'}^{(n-1)}} \sum_{n'=n+1}^{N} \mathcal{L}^{(n')}(\mathbf{X}^{(out,n')})$$

$$= \nabla_{\mathcal{U}_{r'}^{(n-1)}} \mathcal{L}^{(n)}(\mathbf{X}^{(out,n)}) + \sum_{r=1}^{s} \frac{\partial \mathcal{U}_r^{(n)}}{\partial \mathcal{U}_{r'}^{(n-1)}}^{\top} \times \nabla_{\mathcal{U}_r^{(n)}} \mathcal{L}$$

$$= \nabla_{\mathcal{U}_{r'}^{(n-1)}} \left( \mathcal{L}^{(n)}(\mathbf{X}^{(out,n)}) + \sum_{r=1}^{s} \nabla_{\square}[\mathcal{U}_r^{(n)}]^{\top} \langle\langle \nabla_{\mathcal{U}_r^{(n)}} \mathcal{L} \rangle\rangle \right)$$

$$= \nabla_{\mathcal{U}_{r'}^{(n-1)}} \Phi^{(n)}(\theta^{(n)}, \mathcal{U}^{(n-1)}, \nabla_{\mathcal{U}^{(n)}} \mathcal{L})$$

$$= \nabla_{\mathcal{U}_{r'}^{(n-1)}} \Phi^{(n)}(\theta^{(n)}, \mathcal{U}^{(n-1)}, \mathcal{G}^{(n)}),$$

where the second chain of equalities is equivalent to (10).

## C   Efficient "Block" Computation of (2-3)

Denote $\widetilde{\mathbf{Q}} = (g(\mathbf{Q}_l))_{l=1}^{L}, \widetilde{\mathbf{K}} = (g(\mathbf{K}_l))_{l=1}^{L}, \mathbf{N} = (\mathbf{R}_l \times \widetilde{\mathbf{Q}}_l)_{l=1}^{L}, \mathbf{D} = (\mathbf{S}_l^{\top} \widetilde{\mathbf{Q}}_l)_{l=1}^{L}$. [18] propose the following algorithm for computation of (2-3). Initialize buffers $\mathrm{cur}\mathbf{R} = \mathbf{0}_{d \times M}, \mathrm{cur}\mathbf{S} = \mathbf{0}_M$, iterate over $l = 1, \ldots, L$ and compute

$$\mathrm{cur}\mathbf{R} := \mathrm{cur}\mathbf{R} + \mathbf{V}_l \times \widetilde{\mathbf{K}}_l^{\top};$$
$$\mathrm{cur}\mathbf{S} := \mathrm{cur}\mathbf{S} + \widetilde{\mathbf{K}}_l;$$
$$\mathbf{N}_l := \mathrm{cur}\mathbf{R} \times \widetilde{\mathbf{Q}}_l;$$
$$\mathbf{D}_l := \mathrm{cur}\mathbf{S}^{\top} \times \widetilde{\mathbf{Q}}_l;$$
$$\mathbf{Y}_l := \mathbf{N}_l / \mathbf{D}_l.$$

This way, the 3D tensor $\mathbf{R} \in \mathbb{R}^{L \times d \times M}$ is not stored in memory explicitly, resulting in $O(L)$ time and $O(L(d + M) + dM)$ memory complexity. In order to have the same memory consumption during back-propagation, [18] propose the following routine. Keep buffers $\mathrm{cur}\mathbf{R}, \mathrm{cur}\mathbf{S}$ as the result of forward pass, and initialize gradient buffers $\mathrm{grad}\mathbf{R} = \mathbf{0}_{d \times M}, \mathrm{grad}\mathbf{S} = \mathbf{0}_M$. Assuming that $\nabla_{\mathbf{N}} \mathcal{L} \in \mathbb{R}^{L \times d}, \nabla_{\mathbf{D}} \mathcal{L} \in \mathbb{R}^{L}$ are computed using automatic differentiation, iterate in a backward direction $l = L, \ldots, 1$ and compute

$$\nabla_{\widetilde{\mathbf{Q}}_l} \mathcal{L} := (\nabla_{\mathbf{D}_l} \mathcal{L}) \cdot \mathrm{cur}\mathbf{S} + \mathrm{cur}\mathbf{R}^{\top} \times \nabla_{\mathbf{N}_l} \mathcal{L};$$
$$\mathrm{cur}\mathbf{R} := \mathrm{cur}\mathbf{R} - \mathbf{V}_l \times \widetilde{\mathbf{K}}_l^{\top};$$
$$\mathrm{cur}\mathbf{S} := \mathrm{cur}\mathbf{S} - \widetilde{\mathbf{K}}_l;$$
$$\mathrm{grad}\mathbf{R} := \mathrm{grad}\mathbf{R} + (\nabla_{\mathbf{N}_l} \mathcal{L}) \times \widetilde{\mathbf{Q}}_l^{\top};$$
$$\mathrm{grad}\mathbf{S} := \mathrm{grad}\mathbf{S} + (\nabla_{\mathbf{D}_l} \mathcal{L}) \cdot \widetilde{\mathbf{Q}}_l;$$
$$\nabla_{\mathbf{V}_l} \mathcal{L} := \mathrm{grad}\mathbf{R} \times \widetilde{\mathbf{K}}_l;$$
$$\nabla_{\widetilde{\mathbf{K}}_l} \mathcal{L} := \mathrm{grad}\mathbf{R}^{\top} \times \mathbf{V}_l.$$

In practice, the described algorithm works slowly when implemented in pure PyTorch, because $l$ is iterated one-by-one: [18] use low-level CUDA extensions to make the algorithm practical. Instead, we propose a "block" version, when we iterate through blocks of $l$ of a small size $\mathcal{C}$ (we use $\mathcal{C} = 64$). In each block we use explicit prefix sums on inputs of length $\mathcal{C}$ to find $\mathbf{Y}_{l:l+\mathcal{C}-1}$, using the maintained front $\mathrm{cur}\mathbf{R}, \mathrm{cur}\mathbf{S}$. The formal algorithm is as follows. Initialize buffers $\mathrm{cur}\mathbf{R} = \mathbf{0}_{d \times M}, \mathrm{cur}\mathbf{S} = \mathbf{0}_M$. For simplicity assuming that $\mathcal{C}$ divides $L$ (extension for an opposite case is straightforward), iterate over $l = 1, \mathcal{C} + 1, \ldots, L - \mathcal{C} + 1$ and compute

$$\mathrm{block}\mathbf{R} := \mathrm{PS}((\mathbf{V}_{l+l'-1} \times \widetilde{\mathbf{K}}_{l+l'-1}^\top)_{l'=1}^{\mathcal{C}}); \tag{16}$$

$$\mathrm{block}\mathbf{R} := (\mathrm{cur}\mathbf{R} + \mathrm{block}\mathbf{R}_{l'})_{l'=1}^{\mathcal{C}};$$

$$\mathrm{block}\mathbf{S} := \mathrm{PS}((\widetilde{\mathbf{K}}_{l+l'-1})_{l'=1}^{\mathcal{C}}); \tag{17}$$

$$\mathrm{block}\mathbf{S} := (\mathrm{cur}\mathbf{S} + \mathrm{block}\mathbf{S}_{l'})_{l'=1}^{\mathcal{C}};$$

$$\mathrm{cur}\mathbf{R} := \mathrm{block}\mathbf{R}_{\mathcal{C}};$$

$$\mathrm{cur}\mathbf{S} := \mathrm{block}\mathbf{S}_{\mathcal{C}};$$

$$\mathbf{N}_{l:l+\mathcal{C}-1} := (\mathrm{block}\mathbf{R}_{l'} \times \widetilde{\mathbf{Q}}_{l+l'-1})_{l'=1}^{\mathcal{C}};$$

$$\mathbf{D}_{l:l+\mathcal{C}-1} := (\mathrm{block}\mathbf{S}_{l'}^\top \times \widetilde{\mathbf{Q}}_{l+l'-1})_{l'=1}^{\mathcal{C}};$$

$$\mathbf{Y}_{l:l+\mathcal{C}-1} := (\mathbf{N}_{l+l'-1}/\mathbf{D}_{l+l'-1})_{l'=1}^{\mathcal{C}}.$$

In the "block" version, the number of outer sequential iterations is reduced to $L/\mathcal{C}$, resulting in $O((L/\mathcal{C}) \log \mathcal{C})$ parallel time complexity, when the logarithmic parallel algorithm is used to compute prefix sums (16,17). In our experiments, we use `torch.cumsum` to compute (16,17), which works fast in practice. The memory complexity of the algorithm is $O(L(d+M) + \mathcal{C}dM)$, where the second term is for storing $\mathrm{block}\mathbf{R}$. Assuming that $\mathcal{C}$ is a small constant ($\mathcal{C} = O(1)$), we conclude that the "block" version has $O(L(d+M) + dM)$ memory and $O(L)$ time complexity – same as the algorithm of [18]. As for hidden constants in complexity estimates, the constant inside $O(L)$ time complexity is reduced at the cost of increasing constant of the "small" $dM$ term in the memory complexity (when $d, M \ll L$), making the "block" iterative algorithm a practical choice for computing (2-3).

We further show how to back-propagate through (2-3) in $O((L/\mathcal{C}) \log \mathcal{C})$ time and $O(L(d+M) + \mathcal{C}dM)$ memory. Again, keep buffers $\mathrm{cur}\mathbf{R}, \mathrm{cur}\mathbf{S}$ as the result of forward pass, and initialize gradient buffers $\mathrm{grad}\mathbf{R} = \mathbf{0}_{d \times M}, \mathrm{grad}\mathbf{S} = \mathbf{0}_M$. Assuming that $\nabla_\mathbf{N}\mathcal{L} \in \mathbb{R}^{L \times d}, \nabla_\mathbf{D}\mathcal{L} \in \mathbb{R}^L$ are computed using automatic differentiation, iterate in a backward direction $l = L - \mathcal{C} + 1, L - 2\mathcal{C} + 1, \ldots, 1$ and compute

$$\mathrm{cur}\mathbf{R} := \mathrm{cur}\mathbf{R} - \sum_{l'=l}^{l+\mathcal{C}-1} \mathbf{V}_{l'} \times \widetilde{\mathbf{K}}_{l'}^\top;$$

$$\mathrm{cur}\mathbf{S} := \mathrm{cur}\mathbf{S} - \sum_{l'=l}^{l+\mathcal{C}-1} \widetilde{\mathbf{K}}_{l'};$$

$$\mathrm{block}\mathbf{R} := \mathrm{PS}((\mathbf{V}_{l+l'-1} \times \widetilde{\mathbf{K}}_{l+l'-1}^\top)_{l'=1}^{\mathcal{C}});$$

$$\mathrm{block}\mathbf{R} := (\mathrm{cur}\mathbf{R} + \mathrm{block}\mathbf{R}_{l'})_{l'=1}^{\mathcal{C}};$$

$$\mathrm{block}\mathbf{S} := \mathrm{PS}((\widetilde{\mathbf{K}}_{l+l'-1})_{l'=1}^{\mathcal{C}});$$

$$\mathrm{block}\mathbf{S} := (\mathrm{cur}\mathbf{S} + \mathrm{block}\mathbf{S}_{l'})_{l'=1}^{\mathcal{C}};$$

$$\nabla_{\widetilde{\mathbf{Q}}_{l:l+\mathcal{C}-1}}\mathcal{L} := ((\nabla_{\mathbf{D}_{l+l'-1}}\mathcal{L}) \cdot \mathrm{block}\mathbf{S}_{l'} + \mathrm{cur}\mathbf{R}_{l'}^\top \times \nabla_{\mathbf{N}_{l+l'-1}}\mathcal{L})_{l'=1}^{\mathcal{C}};$$

$$\mathrm{grad}\mathbf{R} := \mathrm{grad}\mathbf{R} + \sum_{l'=l}^{l+\mathcal{C}-1} (\nabla_{\mathbf{N}_{l'}}\mathcal{L}) \times \widetilde{\mathbf{Q}}_{l'}^\top;$$

$$\mathrm{grad}\mathbf{S} := \mathrm{grad}\mathbf{S} + \sum_{l'=l}^{l+\mathcal{C}-1} (\nabla_{\mathbf{D}_{l'}}\mathcal{L}) \cdot \widetilde{\mathbf{Q}}_{l'};$$

$$\mathrm{blockgrad}\mathbf{R} := \mathrm{PS}(((\nabla_{\mathbf{N}_{l+l'-1}}\mathcal{L}) \times \widetilde{\mathbf{Q}}_{l+l'-1}^\top)_{l'=1}^{\mathcal{C}});$$

$$\mathrm{blockgrad}\mathbf{R} := (\mathrm{grad}\mathbf{R} - \mathrm{blockgrad}\mathbf{R}_{l'})_{l'=1}^{\mathcal{C}};$$

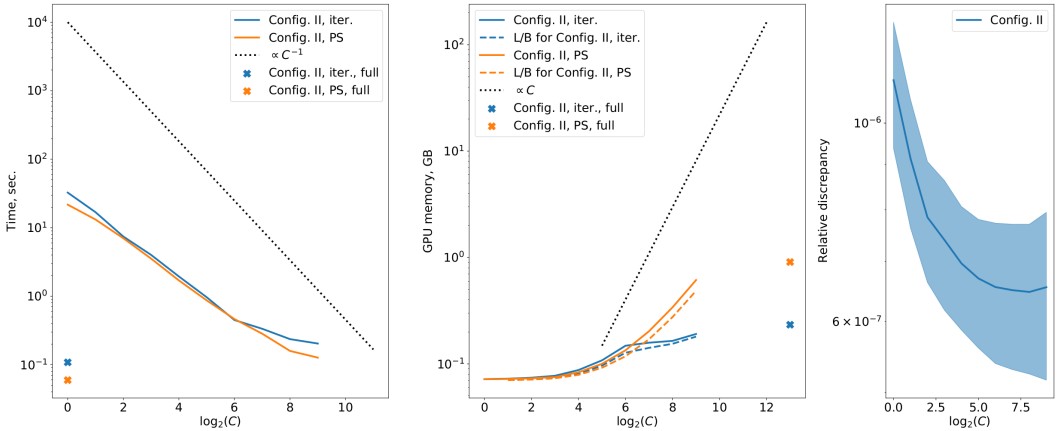

Figure 5: Version of Figure 2 for configuration I.

$$\mathrm{blockgrad}\mathbf{S} := \mathrm{PS}(((\nabla_{\mathbf{D}_{l+l'-1}}\mathcal{L}) \cdot \widetilde{\mathbf{Q}}_{l+l'-1})_{l'=1}^{\mathcal{C}});$$

$$\mathrm{blockgrad}\mathbf{S} := (\mathrm{grad}\mathbf{S} - \mathrm{grad}\mathbf{S}_{l'})_{l'=1}^{\mathcal{C}};$$

$$\nabla_{\mathbf{V}_{l:l+\mathcal{C}-1}}\mathcal{L} := (\mathrm{blockgrad}\mathbf{R}_{l'} \times \widetilde{\mathbf{K}}_{l+l'-1})_{l'=1}^{\mathcal{C}};$$

$$\nabla_{\widetilde{\mathbf{K}}_{l:l+\mathcal{C}-1}}\mathcal{L} := (\mathrm{blockgrad}\mathbf{R}_{l'}^{\top} \times \mathbf{V}_{l+l'-1})_{l'=1}^{\mathcal{C}}.$$

Finally, it's easy to see how to use both one-to-one and "block" iterative computation as part of Algorithm 1 to compute the update (7-8). For that, when doing a forward computation for some $n, r$, initialize $\mathrm{cur}\mathbf{R}, \mathrm{cur}\mathbf{S}$ from corresponding subvectors of $\mathbf{U}_{B_n-1}^{(r-1,n-1)}$, with the rest of the algorithm unchanged. Similarly, during a backward pass for some $n, r$, initialize $\mathrm{grad}\mathbf{R}, \mathrm{grad}\mathbf{S}$ from corresponding subvectors of $\mathcal{G}^{(n)}$ and leave the rest of the iterative back-propagation algorithm unchanged.

## D    Additional Experimental Details

We use 200K, 100K, 200K SGD iterations in the Copying Task, Penn Treebank, Enwik8 setups respectively. We use Adam optimizer [19] with $\beta_1 = 0.9, \beta_2 = 0.999$ (default configuration used in PyTorch). For the Copying task, we train with a learning rate $10^{-3}$ for 130K iterations and then decrease the learning rate to $10^{-4}$. We use a fixed learning rate of $10^{-4}$ and $2 \times 10^{-4}$ in Penn Treebank and Enwik8 experiments respectively.

Figure 5 is a version of Figure 2 for the configuration II. We draw the same conclusions to those reported in the main text. Figure 6 is a bigger version of Figure 4 from the main text. Figure 7 reports additional experimental results: Bits Per Character for the Copying Task and train set learning curves for Penn Treebank and Enwik8. Figure 8 is a version of Figure 4, showing a difference $(-)$ between curves and the "Full" curve. We observe that memory-efficient algorithms result in a negligible loss of performance, which we attribute to numerical effects, accumulating with many thousands of iterations.

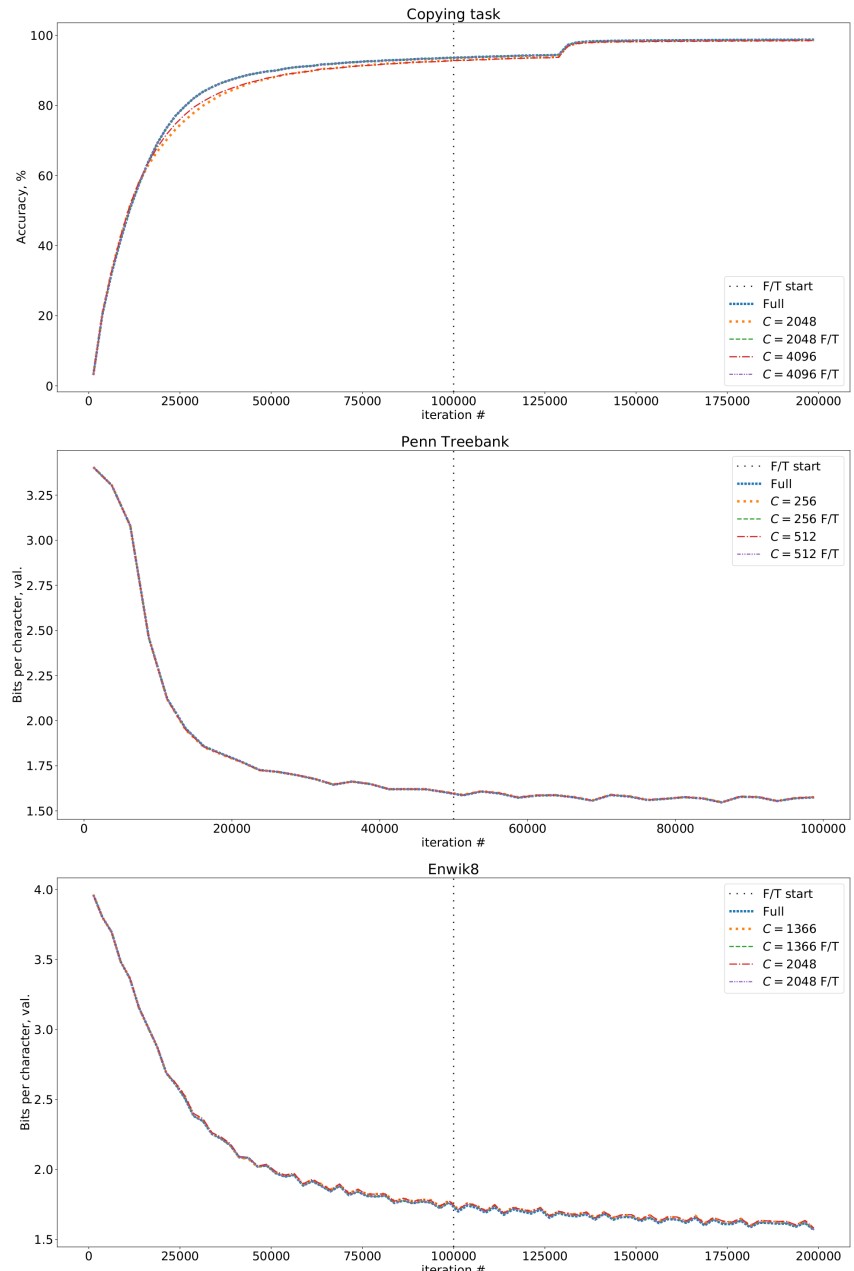

Figure 6: Bigger version of Figure 4.

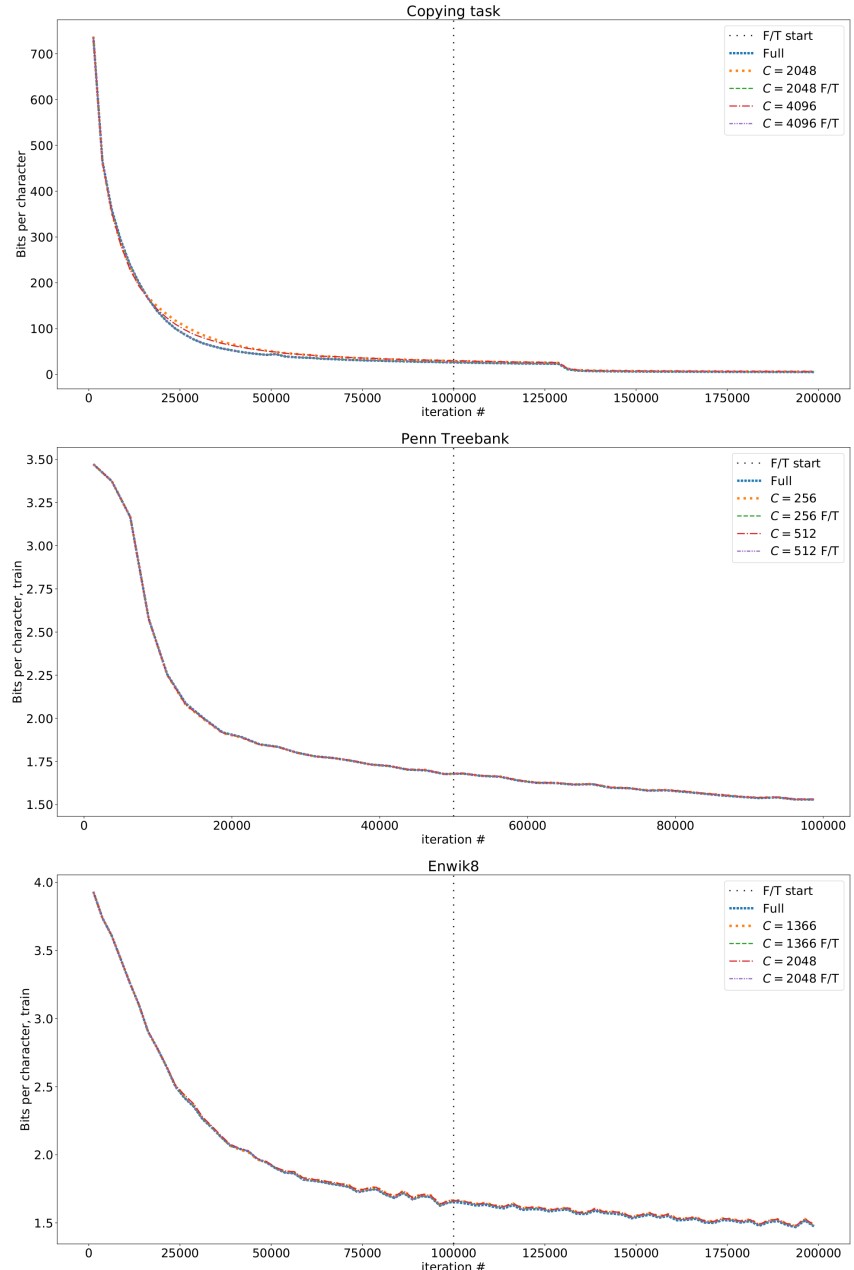

Figure 7: Bits-per-character learning curve for the Copying task and train-set learning curves for language modelling on Penn Treebank and Enwik8 respectively.

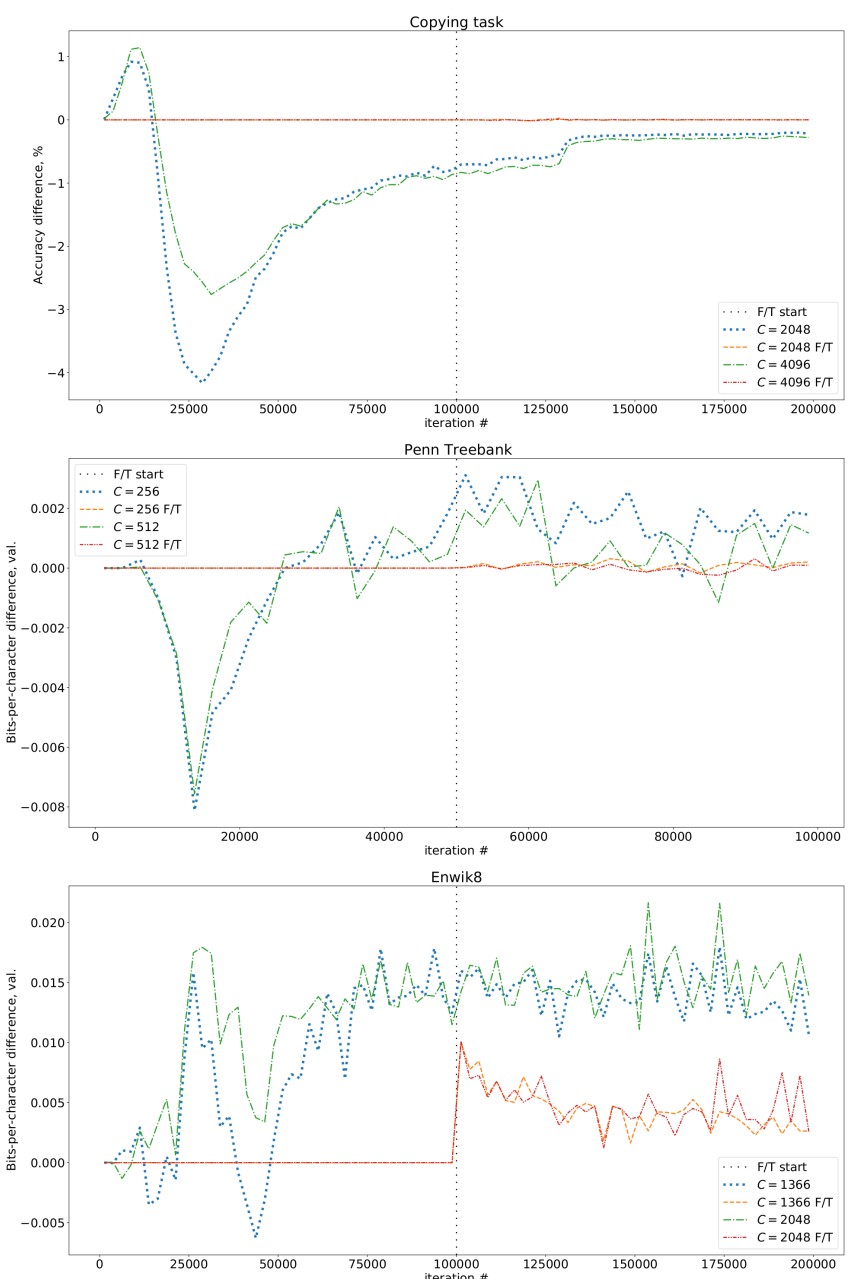

Figure 8: A difference (−) between curves and the "Full" curve from Figure 4.