# OpenReview forum: "Sub-Linear Memory: How to Make Performers SLiM"
_NeurIPS.cc/2021/Conference — NeurIPS 2021 Poster_

### Official Review · Reviewer_NcB2 · 2021-06-30

**Rating:** 7
**Confidence:** 3

**Summary:**

The paper studies memory-time tradeoff and complexity analysis of Performers and proposes an algorithm for memory-efficient back-propagation through a Performer. The experiments highlight the tradeoff between time and memory consumption and the backward-compatibility of the algorithm

**Limitations And Societal Impact:**

Yes

**Main Review:**

The paper studies an interesting problem and provided through analysis of the memory-time tradeoff in training/finetuning Performer models. It discusses interesting implications on training models on edge and low-memory devices and highlights the backward-compatible nature of the algorithm.

Some comments/questions:

The tasks studied are fairly limited in number and type. Did the author consider other tasks  (e.g. tasks from the long arena benchmark)?

The paper cites the competitive performance of Performers among other methods for long sequence modelling and opts to not report tasks performance numbers. It would be useful to report that, especially as relates to the discussion on gradient discrepancy

In 4.1, the paper states that the memory scales slower than O(C), can you please elaborate on why you think this is the case?

While the paper acknowledges the limitation of the work relying on a specific feature of Performers, it would be useful to discuss if any of the findings could be extended to other efficient Transformers architectures

**Time Spent Reviewing:**

1

---

> ### Author Response · Authors · 2021-08-10
> **Response to Reviewer NcB2**
>
> Thank you! We address the reviewer's questions below.
>
> > Did the author consider other tasks (e.g. tasks from the long arena benchmark)?
>
> Since our algorithm is arithmetically equivalent to Performer, and Performer was evaluated in the long range arena, we believe that a duplicate comparison is redundant and opt for other setups instead.
>
> > It would be useful to report that, especially as relates to the discussion on gradient discrepancy.
>
> Regarding Performer's evaluation in other tasks, we refer to the corresponding literature (e.g. [1]). Regarding tasks considered in our paper, Performer's evaluation is illustrated in Figure 4 ("full") -- i.e. our algorithm has an almost indistinguishable performance. This is in line with a negligible reported gradient discrepancy (Figure 2).
>
> > Can you please elaborate on why you think this is the case?
>
> We relate that to details of Pytorch's instruments for memory monitoring (torch.cuda.max_memory_allocated) and/or Pytorch's internal work with memory, e.g. it looks like PyTorch doesn’t release some of the unused memory, which results in higher memory consumption for a small C.
>
> > It would be useful to discuss if any of the findings could be extended to other efficient Transformers architectures.
>
> Our algorithm can be applied to all Performers [1], linear Transformers [2] and parallelizable recurrent neural networks [3], since they employ prefix sum operation for temporal propagation, which is a crucial condition. We believe this is a significant contribution: while relatively young, Performer [1] and linear Transformer [2] architectures have already been adopted in many applications (each paper already has more than 100 citations) [see response to Reviewer Bx7L above for further references].
>
> [1] Krzysztof Choromanski, Valerii Likhosherstov, David Dohan, Xingyou Song, Andreea Gane, 325 Tamas Sarlos, Peter Hawkins, Jared Davis, Afroz Mohiuddin, Lukasz Kaiser, David Belanger, Lucy Colwell, and Adrian Weller. Rethinking attention with Performers. In International Conference on Learning Representations, 2021.
>
> [2] Katharopoulos, A., Vyas, A., Pappas, N., & Fleuret, F. (2020, November). Transformers are rnns: Fast autoregressive transformers with linear attention. In International Conference on Machine Learning (pp. 5156-5165). PMLR.
>
> [3] Eric Martin and Chris Cundy. Parallelizing linear recurrent neural nets over sequence length. In International Conference on Learning Representations, 2018.

---

### Official Review · Reviewer_s7XB · 2021-07-14

**Rating:** 6
**Confidence:** 3

**Summary:**

The authors propose a low-memory gradient computation algorithm for the linear transformer architecture. Their algorithm allows the user to control the trade-off between computational efficiency and memory by specifying how many tokens of the input sequence should be processed at once in the forward pass. At the extreme, the memory consumption can be reduced to a single token (+model parameters +some overhead)
They empirically demonstrate that a) the obtained gradient is correct even considered finite-precision arithmetic, b) the actual memory consumption of their implementation aligns with the claims of their analysis, c) the trade-off between memory and compute can be dynamically adjusted during training to allow for fine-tuning on low-memory devices.


**Limitations And Societal Impact:**

yes

**Main Review:**

I agree with the authors that memory limitations are an important concern and constraint for training of large-scale machine learning models. Trading memory from computational time allows you to train a model on a smaller machine, with the caveat that you potentially have to wait for a long time to get your results, but it is possible.

Overall I think the paper is well written and the proposed implementation is clearly described. I am not an expert on transformer architectures so I leave the judgement of novelty and significance to other reviewers.  The only concern I have is that the approach is quite niche and I am not sure how transferrable and relevant this algorithm is for state-of-the-art architectures.

Some questions I have are
-	You need to be able to change the order of the nested loop in computing (5). How general is this implementation tick, can it be extended from linear to more sophisticated transformer architectures?
-	For larger batch sizes a common approach to reduce memory is to do gradient accumulation. Do you need to load full sequences to do gradient accumulation or could your approach also be combined with larger batches and gradient accumulation? You would probably have issues with reading data from storage in chunks I suppose.
-	Are you planning to make your code publicly available or integrate it into existing transformer implementations?
-	You are running your experiments on a P100 GPU. I would be curious to know the utilization of the device for different values of C to get an idea of how much sequential nature of your approach and the memory constraints are limiting your ability to fully utilize the GPU.
-	You mention that this approach would allow for fine-tuning of transformer models on devices such as smartphones or microcontrollers. But then your experiments are on a P100 GPU which seems quite a bit more powerful. Do you think these experiments could actually be done on a smartphone?


**Time Spent Reviewing:**

4

---

> ### Author Response · Authors · 2021-08-10
> **Response to Reviewer s7XB**
>
> Thank you for your valuable feedback, and noting the importance of memory limitations! Performer [1] and linear Transformer [2] architectures were introduced recently, yet have already been adopted in many applications (each paper already has more than 100 citations). Hence, we believe that our proposed algorithm is not niche and has a wide range of applications [see references in our response to Reviewer Bx7L]. We address the reviewer's other questions in order below:
>
> 1. Representation (5) is crucial for our algorithm, since it relies on the fact that temporal information propagation is done through prefix sum. This representation holds for all Performers [1], linear Transformers [2] and parallelizable recurrent neural networks [3]. As we mentioned just above, these architectures appear in many applications, so we believe our contribution is valuable. We leave possible extensions of our algorithm to future work.
>
> 2. Our approach can be combined with gradient accumulation -- one can just use our algorithm to compute gradients for each input string and accumulate them over the batch. We believe that input batch storage is not typically the bottleneck of gradient computation, since one input token requires around 1 or 2 bytes, whereas the intermediate representation in the Transformer is a vector of X floats, where X = 512, 768, … is the hidden size of the Transformer, and there are many such representations.
>
> 3. Yes, all our code, as we supplied in the supplementary materials, will be made publicly available.
>
> 4. We report GPU memory consumption and running time of the algorithm depending on C on Figure 2 (left, middle). More sequential computations (smaller C) result in less memory consumption according to this Figure. Note that this result is implementation and platform-specific, while we also provide a comprehensive theoretical analysis which is implementation-agnostic.
>
> 5. Our paper is dedicated to a mathematical description and analysis of a newly proposed algorithm, and reproducible empirical validation in a well-known setup. For this reason, we used a standard GPU. However, our analysis is generic and applicable to any computational units, so our algorithm should lead to memory improvements on smartphones as well. We leave evaluation on specific hardware (smartphones, etc.) for future work.
>
> [1] Krzysztof Choromanski, Valerii Likhosherstov, David Dohan, Xingyou Song, Andreea Gane, 325 Tamas Sarlos, Peter Hawkins, Jared Davis, Afroz Mohiuddin, Lukasz Kaiser, David Belanger, Lucy Colwell, and Adrian Weller. Rethinking attention with Performers. In International Conference on Learning Representations, 2021.
>
> [2] Katharopoulos, A., Vyas, A., Pappas, N., & Fleuret, F. (2020, November). Transformers are RNNs: Fast autoregressive transformers with linear attention. In International Conference on Machine Learning (pp. 5156-5165). PMLR.
>
> [3] Eric Martin and Chris Cundy. Parallelizing linear recurrent neural nets over sequence length. In International Conference on Learning Representations, 2018.

---

> > ### Comment · Reviewer_s7XB · 2021-08-25
> > **Thanks**
> >
> > I thank the authors for the response to my questions.
> >
> > I understand that the trick is specific to prefix sums. I still think it would be nice if the authors could add a short discussion on this point and mentioning potential extensions and limitations of how this approach is applicable to modified transformer architectures.
> >
> > Just to clarify - in point 4 I was interested in the utilization of the GPU rather than the running time of the algorithm, to get an indication for how much room there is for improvement on the computational side.
> >
> > I have also read the other reviews and responses. I am on board accepting the paper.

---

### Official Review · Reviewer_M3kX · 2021-07-17

**Rating:** 6
**Confidence:** 4

**Summary:**

This work proposes a modification of linear self-attention that allows to trade off parallel execution speed for memory usage. Authors provide extensive comparison of their method with baseline approaches in several configurations and on multiple datasets.

**Limitations And Societal Impact:**

Yes

**Main Review:**

Strengths:
1) A simple but effective approach to memory usage reduction for efficient Transformers. Authors clearly demonstrate the advantages of their approach both theoretically and in experiments.
2) In general, the paper is well-written and supplied with helpful illustrations.
3) The work addresses a problem of memory efficiency for Transformers, which is quite important to the research community.

Weaknesses:
1) Currently, the paper is quite hard to understand due to the heavy usage of custom notation. Describing the core idea in a simpler way without the notation can considerably improve the flow of sections 2 and 3.
2) Though the benchmarks are reasonable in general, they deviate from standard Transformer architectures in two ways. First, the experiments use no more than 3 Transformer layers, and most of today's architectures use at least 6 of them. Second, the vocabulary size of 256 is used very rarely (e.g., for byte-level models without BPE). As a result, it is not entirely clear whether the reported gains will translate into the same improvements in a more practical setting.

Typos:
L66: "subqudratic" -> "subquadratic"

**Time Spent Reviewing:**

3

---

> ### Author Response · Authors · 2021-08-10
> **Response to Reviewer M3kX**
>
> Thank you for your valuable feedback and kind words in "Strengths"! We address points indicated as "weaknesses" below:
>
> 1. We will include a high-level description of the main idea in a paragraph called "High-level description of the algorithm" at the end of Section 3.1. There we will mention that the algorithm iterates over the sequence and only maintains a front of current prefix sum values, thus allowing a substantial memory improvement. The backward pass is implemented similarly: current prefix sums along with their gradients are maintained in a dynamic programming fashion, but the iteration proceeds in a backward direction.
> We will do additional readability improvements and we are happy to see any other comments/suggestions regarding that aspect.
>
> 2. Our proposed algorithm for gradient computation is arithmetically equivalent to full-length back-propagation in Performer, and Performer is thoroughly evaluated in [1,2]. In our paper, we show how the benefits of Performers can be obtained with significantly less computational resources (1 GPU) -- an important consideration for enabling the broad academic community to enjoy the use of these models. Hence we evaluate our approach in settings accessible for a wider audience with less access to compute resources (e.g. PhD students). We believe these considerations address the reviewer's concern. 3-layer Transformers are used in practice for smaller-size datasets, e.g. in [3] for a Penn Treebank experiment.
>
> Thank you for catching the typo in L66. We will fix this and all other suggestions from reviewers in the final version.
>
> [1] Krzysztof Choromanski, Valerii Likhosherstov, David Dohan, Xingyou Song, Andreea Gane, 325 Tamas Sarlos, Peter Hawkins, Jared Davis, Afroz Mohiuddin, Lukasz Kaiser, David Belanger, Lucy Colwell, and Adrian Weller. Rethinking attention with Performers. In International Conference on Learning Representations, 2021.
>
> [2] Yi Tay, Mostafa Dehghani, Samira Abnar, Yikang Shen, Dara Bahri, Philip Pham, Jinfeng Rao, Liu Yang, Sebastian Ruder, and Donald Metzler. Long range arena : A benchmark for efficient transformers. In International Conference on Learning Representations, 2021.
>
> [3] Ma, X., Zhang, P., Zhang, S., Duan, N., Hou, Y., Zhou, M., & Song, D. (2019). A tensorized transformer for language modeling. Advances in Neural Information Processing Systems, 32, 2232-2242.

---

> > ### Comment · Reviewer_M3kX · 2021-08-26
> > **Thank you for the response!**
> >
> > Thank you for a detailed response! I appreciate the planned steps towards improving the clarity of the paper, and if the proposed changes are added to the work, one of my major concerns regarding the paper will be resolved. As a result, I've changed my rating from 5 to 6.

---

### Official Review · Reviewer_Bx7L · 2021-07-19

**Rating:** 7
**Confidence:** 4

**Summary:**

This paper studies a time and memory trade-off for training Performers with the linear attention mechanism. The goal of the paper is to study this trade-off under memory constrained setting, and propose a memory efficient performer training algorithm at the cost of runtime. The contributions of the paper are:

1. Given a sequence of length L, the authors proposed a dynamic programming algorithm which can process the sequence as a collection of segments of length C (the segments are coupled). This reduce the training memory for performer from O(L) to O(C) without any approximation. The intuition here is that the information across segments only flow through certain partial sums over tensors associated with each token in the sequence.

2. The authors analyzed the memory / time trade-off in the proposed algorithm (including extremal cases).

3. Empirically, the authors shows that 1) by controlling the segment length C, the proposed algorithm indeed present a trade off between runtime and memory; 2) The gradient for performer models only have minimal / negligible difference when using the conventional and the proposed algorithm; 3) A large performer model pretrained with the conventional algorithm can be finetuned via the proposed algorithm with small segment lengths; this achieves accuracy matching the conventional algorithm with less memory.



**Ethical Concerns:**

I do not see anything improper or having ethical issues with this paper.

**Limitations And Societal Impact:**

I think the attached checklist answers the questions adequately. The overhead / limitation of the method is discussed in the conclusion section.

**Main Review:**

Originality:

The intuition and ideas in the proposed algorithm is novel when applies in reduced the memory of performer training.

Quality:

In this paper, the proposed algorithm can reduce the performer training memory from O(L) to O(C) where L is the sequence length and C is the segment length (C < L). The value of C present a knob to exploit the time and memory trade-off in this algorithm. When C is small, the algorithm can reduce performer training memory significantly. Here are a few suggestions to further enhance the paper (and I am willing to raise the score if the questions are resolved and incorporated into writing)

1. The derivation and discussion in the paper are mostly focusing on the one-directional casual language modeling setting. In the modern NLP pipelines, the bi-directional masked language modeling setting is another equally important one. Does the trade-off discussion and complexity still applies to bi-directional setting?

2. I saw the memory saving with small C is only ~2X compared to conventional performer training algorithm (Table 3)? It would be great if the authors could explicitly decompose the memory consumption so that practitioners can know where to attach to make it more savings when implemented in lower level (instead of using the coarse grain pytorch or other frameworks)?

3. The exact attention compute in conventional transformers (vs performers) can be approximated with performers using tailored feature mapping functions g(*). E.g. the exp kernel to compute attention score can be approximated with say linear features (e.g. random fourier features or Nystrom approximation). For the conventional transformer MHA via exp kernel, does the proposed approach with a tailored g(*) provide matching accuracy as conventional MHA while significantly save the memory?

Clarity:

I think the paper is well-written in the high level. I have one major comments and a few minor comments.

Major: The discussion in sec 3.3 can be further improved to allow for easier reading. Essentially, the backward algorithm there is just a recursive / iterative to compute \partial L / partial \theta_{n} and \partial L / partial \U_{n} where \theta_{n} is the symbolic model param for the n-th segment and \U_{n} is the partial sum related quantities for the n-th segment. The current discussion is a bit scattered to be digested very quickly. By giving a theorem or algorithm box on the equations to compute \partial L / partial \theta_{n} and \partial L / partial \U_{n} would be useful for clarity.

Minor:
1. Line 163, typo. It should be serial time O(L/C log C)
2. Line 178, curious when saying two forward, does it just mean a full recompute of forward activations for the current segment?


Significance: This paper can have positive impact on memory efficient implementation for performers. It can also potentially lead to wider application into more transformer variants.

**Time Spent Reviewing:**

6 hours

---

> ### Author Response · Authors · 2021-08-10
> **Response to Reviewer Bx7L**
>
> We thank the reviewer for the valuable feedback! Below are responses to the reviewer's suggestions:
>
> 1. The idea behind the algorithm proposed in the paper is that we can maintain a "front" of prefix sum values or their gradients and update this front as we move forward or backward. This idea cannot be extended directly to the bidirectional case, since no such notion of front can be introduced in that case. We believe, however, that our contribution is still significant, since the unidirectional case is very important and arises in many applications, including language modelling, image generation, sequence-to-sequence modelling, etc. We will add a discussion of the bidirectional case into the "Limitations" paragraph, thanks for the suggestion!
>
> 2. We checked that only loading Transformer weights into GPU results in ~0.0661 Gb and ~0.2610 Gb memory consumption for PTB and ENW experiments respectively. This gives memory savings of the gradient computation (not the parameter storage) of a factor
>
> (0.1834 - 0.0661) / (0.1077 - 0.0661) ~ 2.82x for PTB, and
> (0.8049 - 0.2610) / (0.4276 - 0.2610) ~ 3.26x for ENW,
>
> which is substantial. We will add these results to the final revision, thank you so much for pointing this out! Still, these numbers are implementation specific, and the maximal possible theoretical memory improvements are derived in Section 3.4 -- up to the memory required to process a single sequence element, plus a small addition. We believe these strict theoretical bounds should be taken into account by practitioners aiming at the best possible memory improvement.
>
> 3. The suggested approximations are extensively discussed in [1] (for random features) and in [2] (for Nystrom-based approximation), with theoretical results and rigorous empirical evaluations. We focus on the computational aspects and tradeoffs of Performer models from [1] -- this is a broad class of models which have already been shown to be practically important across many domains, including text, image, protein generation [1,3], neural machine translation [2], music generation [4]. Hence, we believe that the discussions on self-attention approximation are outside the scope of our paper. Since our algorithms are arithmetically equivalent to the full-length gradient computation in Performer, all results from [1] should be preserved by our proposed algorithm.
>
> We will incorporate the reviewer's other major and minor comments into the final revision, thank you. Regarding the minor comment 2, we mean that for each segment, we do one forward pass during the forward iteration and one during the backward iteration to construct a symbolic graph for \Phi.
>
> [1] Krzysztof Choromanski, Valerii Likhosherstov, David Dohan, Xingyou Song, Andreea Gane, 325 Tamas Sarlos, Peter Hawkins, Jared Davis, Afroz Mohiuddin, Lukasz Kaiser, David Belanger, Lucy Colwell, and Adrian Weller. Rethinking attention with Performers. In International Conference on Learning Representations, 2021.
>
> [2] Xiong, Y., Zeng, Z., Chakraborty, R., Tan, M., Fung, G., Li, Y., & Singh, V. (2021). Nystromformer: A Nystrom-Based Algorithm for Approximating Self-Attention. arXiv preprint arXiv:2102.03902.
>
> [3] Jungo Kasai, Hao Peng, Yizhe Zhang, Dani Yogatama, Gabriel Ilharco, Nikolaos Pappas, Yi Mao, Weizhu Chen, Noah A. Smith. Finetuning Pretrained Transformers into RNNs. arXiv preprint arXiv:2103.13076.
>
> [4] Gaëtan Hadjeres, Léopold Crestel. The Piano Inpainting Application. arXiv preprint arXiv:2107.05944.

---

### Decision · Program_Chairs · 2021-09-27

**Decision:**

Accept (Poster)

**Comment:**

The reviewers have reached a consensus in favor of accepting this paper. I concur with this consensus. I expect that the proposed changes suggested in the author response would improve the clarity of the final version of the paper.